

# Understanding aerosol-cloud interactions in the development of orographic cumulus congestus during IPHEx

Yajuan Duan[1], Markus D. Petters[2], and Ana P. Barros[1\$]

[1]Department of Civil and Environmental Engineering, Duke University, Durham, NC, USA

5  [2]Department of Marine Earth and Atmospheric Sciences, North Carolina State University, Raleigh, NC, USA

[\$]*Correspondence to*: Ana P. Barros (barros@duke.edu)

**Abstract.** A new cloud parcel model (CPM) including activation, condensation, collision-coalescence, and lateral entrainment processes is presented here to investigate aerosol-cloud interactions (ACI) in cumulus development prior to rainfall onset. The CPM was employed along with ground based radar and surface aerosol measurements to predict the vertical structure of cloud formation at early stages and evaluated against airborne observations of cloud microphysics and thermodynamic conditions during the Integrated Precipitation and Hydrology Experiment (IPHEx) over the Southern Appalachian Mountains. Further, the CPM was applied to explore the space of ACI physical parameters controlling cumulus congestus growth not available from measurements, and to examine how variations in aerosol properties and microphysical processes influence the evolution and thermodynamic state of clouds over complex terrain via sensitivity analysis. Modelling results indicate that aerosol-cloud droplet number concentration (CDNC) closure is achieved optimally to ~ 1.3% of the observations for condensation coefficient ($a_c$) = 0.01 and within 5% for $0.01 < a_c < 0.015$, and the corresponding spectra in the predictions are in good agreement with IPHEx aircraft observations around the same altitude. This is in contrast with larger closure errors and high $a_c$ values reported in previous studies assuming adiabatic conditions. Entrainment is shown to govern the vertical development of clouds and the change of droplet numbers with height, and the sensitivity analysis suggests that entrainment strength and condensation process are mutually compensating to attain aerosol-CDNC closure. Simulated CDNC also exhibits high sensitivity to variations in initial aerosol concentration at cloud base, but weak sensitivity to aerosol hygroscopicity. Exploratory multiple-parcel simulations capture realistic time-scales of vertical development of cumulus congestus (deeper clouds and faster droplet growth). These findings provide new insights into determinant factors of mid-day cumulus congestus formation that can explain a large fraction of warm season rainfall in mountainous regions.

## 1 Introduction

Atmospheric aerosols produced by dramatically increased industrialization and urbanization exert a large impact on the climate system and the hydrological cycle (Koren et al., 2008;Ramanathan et al., 2001;Tao et al., 2012). Aerosols influence the earth-atmosphere system primarily via two mechanisms: a radiative (direct) effect and a microphysical





(indirect) effect (Rosenfeld et al., 2008). The direct effect on the Earth's energy budget occurs via scattering and absorbing of shortwave and longwave radiation in the atmosphere, hence modulating the net radiation and climate (Haywood and Boucher, 2000;Ramanathan et al., 2001). The indirect effect is related to aerosols as cloud condensation nuclei (CCN) or ice nuclei (IN) that alter microphysical properties and consequently affect cloud radiative properties and precipitation efficiency

(Jiang et al., 2008;Lohmann and Feichter, 2005;McFiggans et al., 2006). In particular, an increase in aerosol concentration results in enhanced cloud droplet number concentration (CDNC), smaller average drop size, and increased cloud albedo (Twomey, 1977). Smaller cloud droplets are associated with lower collection and coalescence efficiency, slower drop growth and reduced precipitation, thus leading to longer cloud lifetimes (Albrecht, 1989;Andreae and Rosenfeld, 2008;Khain et al., 2005). Over complex terrain in California and Israel, Givati and Rosenfeld (2004) attributed a reduction in annual

precipitation of 15–25% to air-pollution aerosols from upwind urban areas. Such local effects can translate into large spatial shift in clouds and precipitation in that aerosol-cloud interactions (ACI) inducing suppression of precipitation upwind could give rise to the enhancement of precipitation downwind, thus shifting the spatial distribution of orographic precipitation which can strongly influence the hydrological cycle at local scales as  shown by Muhlbauer and Lohmann (2008).

Observations collected over complex terrain during IPHEx (Integrated Precipitation and Hydrology Experiment;

Barros et al., 2014) provide a great opportunity to investigate ACI in an orographic context of the Southern Appalachian Mountains (SAM). Previous research (Wilson and Barros 2014) showed that seeder-feeder interactions among multilayer clouds generally, and between locally initiated or propagating convective clouds and low-level boundary layer clouds in particular, can increase the intensity of rainfall by one order of magnitude in the SAM and explain the observed peak mid-day peak in rainfall. Thus, the ability to predict the evolution of cloud formation and the vertical structure of droplet size

distribution (DSD) in this region is of paramount interest.

Because of their multiscale nature and complex physics, the representation of physical and chemical processes related to clouds and precipitation in numerical models relies on parameterizations with varying degrees of uncertainties depending on space-time model resolution (Khairoutdinov et al., 2005;Randall et al., 2003). For example, the characteristic time-scale of condensational growth of submicron-size droplets is on the order of 1 ms, and length scales of individual drops

range from μm to cm (Pinsky and Khain, 2002), that is a scale gap of six to nine-orders in magnitude with respect to the spatial resolution of cloud-resolving models (kms). Although detailed 2-D and 3-D models that explicitly resolve cloud formation and microphysical evolution to varying degrees of completeness have been developed for applications in deep convective clouds including both warm- and ice-phase processes (Fan et al., 2009;Leroy et al., 2009), relatively large time steps and coarse spectral resolution of aerosols and cloud droplets are employed for computational efficiency, and these

processes are highly parameterized. Analysis of high resolution (~ 1 km) numerical weather prediction (NWP) simulations in the SAM for various hydrometeorological regimes using different Weather Research and Forecasting (WRF) physical parameterizations concluded that the prediction of cloud development and cloud vertical microphysical structure are inadequate to capture the spatial and temporal resolution of precipitation rate and precipitation microphysics at the ground (Wilson and Barros, 2015, 2017).



An alternative modelling approach to investigate ACI is the cloud parcel model (CPM) that simulates aerosol activation and cloud droplet growth, as well as thermodynamic adaptation of ascending air parcels at μm and ms scales (Abdul-Razzak et al., 1998;Cooper et al., 1997;Flossmann et al., 1985;Jacobson and Turco, 1995;Kerkweg et al., 2003;Nenes et al., 2001;Pinsky and Khain, 2002;Snider et al., 2003). A synthesis of model formulation including spectral

binning strategy, principal physical processes (i.e., condensational growth, collision-coalescence, entrainment), and key aspects of their numerical implementation is presented in Table 1 for CPMs frequently referred to in the peer-reviewed literature. In the past, process studies using CPMs targeted principally aerosol-CDNC closure between model simulations and field observations. For example, Conant et al. (2004) conducted an aerosol-cloud droplet number closure study against observations from NASA's Cirrus Regional Study of Tropical Anvils and Cirrus Layers–Florida Area Cirrus Experiment

(CRYSTAL-FACE) using the adiabatic CPM by Nenes et al. (2001;2002) that solves activation and condensation processes only (see Table 1 for details). Using a condensation coefficient ($a_c$) value of 0.06, they reported that predicted CDNC was on average within 15% of the observed CDNC in adiabatic cloud regions. Fountoukis et al. (2007) used the same CPM as Conant et al. under extremely polluted conditions during the 2004 International Consortium for Atmospheric Research on Transport and Transformation (ICARTT) experiment. They found that the optimal closure of cloud droplet concentrations

was achieved when the condensation coefficient was about 0.06. For marine stratocumulus clouds sampled during the second Aerosol Characterization Experiment (ACE-2), Snider et al. (2003) applied the UWyo parcel model (http://www.atmos.uwyo.edu/~jsnider/parcel/) to simulate condensation processes in adiabatic ascent (see Table 1) and experimented with various condensation coefficients in the range of 0.01–0.81. They hypothesized that the lower CDNC overestimation errors (20 to 30% for $a_c = 0.1$) in their CPM simulations could be mitigated by varying the condensation

coefficient as a function of dry particle size instead of using one value for the entire distributions, but did not actually demonstrate this was the case. The condensation coefficient of water is a key ACI physical parameter in parcel models and has a strong influence on activation and droplet growth by condensation as it expresses the probability that vapour molecules impinge on the water droplet when they strike the air-water interface (McFiggans et al., 2006). Experimental measurements reviewed by Marek and Straub (2001) exhibit a strong inverse relationship between pressure and $a_c$ values ranging from

1000 hPa to 100 hPa and from 0.007 to 0.1, respectively (their Fig. 4). Chodes et al. (1974) measured condensation coefficients in the range of 0.02–0.05 with a mean of 0.033 from measurements of individual droplets grown in a thermal diffusion chamber for four different supersaturations. Ganier et al. (1987) repeated Chodes et al.'s experiments and found that the average condensation coefficient is closer to 0.02 after correcting their supersaturation calculations. Shaw and Lamb (1999) conducted extensive simultaneous measurements of the condensation coefficient and thermal accommodation

coefficients ($a_T$) for individual drops in a levitation cell and reported values for $a_c$ and $a_T$ in the ranges of 0.04–0.1 and 0.1–1 with most probable values of 0.06 and 0.7, respectively. Using adiabatic CPMs with laboratory based condensation coefficients, the resulting errors, mostly due to overestimation, are still well above 10% in aerosol-cloud droplet number closure studies (McFiggans et al., 2006).





Beyond aerosol-CDNC closure, the focus of this study is also on the spatial evolution the droplet spectra as a function of height that determines the vertical microstructure of clouds against airborne observations. For this purpose, a new spectral CPM was developed aiming to replicate independently aircraft microphysical observations during IPHEx, which solves explicitly the cloud microphysics of condensation, collision-coalescence, and lateral entrainment processes.

Numerical experiments were conducted with the objective of elucidating the quantitative impact on cloud formation at early developments of key ACI modelling parameters (e.g., condensation coefficient, entrainment strength), as well as initial conditions (e.g. aerosol properties, thermodynamic conditions in the atmosphere). Surface aerosol measurements sampled during IPHEx and sounding profiles from WRF simulations were used to initialize the parcel model. Predicted cloud droplet spectra and vertical profiles of thermodynamic variables are evaluated against airborne measurements for a cumulus

congestus case-study to elucidate determinant factors in the microphysical evolution of clouds, at early stages in particular. Model sensitivity experiments were conducted to provide insight into possible ranges of major ACI modelling parameters in the SAM, which were not available before or during IPHEx. Multi-parcel simulations were performed to examine the realistic evolution and vertical development of cumulus clouds, which are formed by multiple air parcels rising in succession (Roesner et al., 1990).

The manuscript is organized as follows. The mathematical formulation of the cloud parcel model is briefly described in Sect. 2. Section 3 presents the IPHEx measurements relevant for the modelling study. In Sect. 4, model sensitivity tests against in situ observations are conducted focusing on exploring physically-meaningful ranges of key parameters of ACI and identifying major contributors to cloud formation over the complex terrain of the SAM. Results from multi-parcel simulations highlight the importance of coupled thermodynamics-microphysics to reproduce the realistic

formation of cumulus clouds. Finally, a discussion of the main findings and a brief outlook of ongoing and future research are presented in Sect. 5.

**2 Model description**

To investigate the evolution of cloud droplet spectra originating from aerosol distributions of uniform chemical composition, a new cloud parcel model (hereafter DCPM, or Duke CPM for specificity) was developed to explicitly solve

key cloud microphysical processes (see the last row of Table 1 for details). The model synthesizes well-established theory and physical parameterizations in the literature. In particular, condensation and lateral homogeneous entrainment follow the basic formulations of Pruppacher and Klett (1997) and Seinfeld and Pandis (2006) albeit modified to incorporate the single parameter representation of aerosol hygroscopicity (Petters and Kreidenweis, 2007). The representation of collision-coalescence processes takes into account the variation of collision efficiencies with height (Pinsky et al., 2001), and the

effects of turbulence on drop collision efficiency as per Pinsky et al. (2008). The model discretizes the droplet spectra on a finite number of bins (*nbin*) using a discrete geometric volume-size distribution, spanning a large size range with fewer bins and very fine discretization in the small droplet sizes to improve computational efficiency (Kumar and Ramkrishna,





1996;Prat and Barros, 2007). The characteristic single-particle volumes in adjacent bins are expressed as $v_{i+1} = V_{rat} v_i$, where $V_{rat}$ is a constant volume ratio (Jacobson, 2005). When condensation and coalescence are solved simultaneously, a traditional stationary (time-invariant) grid structure often introduces artificial broadening of the droplet spectrum by reassigning droplets to fixed bins through interpolation, that is numerical diffusion (Cooper et al., 1997;Pinsky and Khain, 2002). To

eliminate numerical diffusion artefacts, a moving grid structure is implemented so that an initial size distribution based on a fixed grid discretization can change with time according to the condensational growth. This approach allows particles in each bin to grow by condensation to their exact transient sizes without partitioning between adjacent size bins. Subsequently, collision and coalescence are resolved on the moving bins that evolve from condensation. The DCPM predicts number and volume concentrations of cloud droplets and interstitial aerosols, liquid water content (LWC), effective drop radius,

reflectivity and other moments of DSD. It also tracks thermodynamic conditions (e.g., supersaturation, temperature, pressure) of the rising air parcel. The flowchart in Fig. 2 graphically describes the key elements and linkages in the parcel model, including microphysical processes, and main inputs and outputs. A detailed description of the formulation of key processes in presented next. A glossary of symbols as well as additional formulae are summarized in Appendix A. The performance of the DCPM was first evaluated by comparing its dependence on different parameters with the results from the

numerical simulations reported by Ghan et al. (2011) as shown in Sect. S1 (see the supplementary material).

### 2.1 Condensation growth with entrainment

The time variation of the parcel's temperature ($T$) can be written as

$$-\frac{dT}{dt} = \frac{gV}{c_p} + \frac{L}{c_p}\frac{dw_v}{dt} + \mu\left[\frac{L}{c_p}(w_v - w_v') + (T - T')\right]V \tag{1}$$

where the first two terms on the right-hand side represent adiabatically cooling of a rising parcel and the third term describes the modulation by entraining ambient dry air. The vertical profiles of ambient temperature ($T'$) and water vapour mixing ratio

($w_v'$) can be interpolated from input sounding data from atmospheric model simulations or radiosonde observations.

The change of the water vapour mixing ratio ($w_v$) in the parcel over time is described by

$$\frac{dw_v}{dt} = -\frac{dw_L}{dt} - \mu(w_v - w_v' + w_L)V \tag{2}$$

The change of the parcel's velocity ($V$) is given by

$$\frac{dV}{dt} = \frac{g}{1+\gamma}\left(\frac{T-T'}{T'} - w_L\right) - \frac{\mu}{1+\gamma}V^2 \tag{3}$$

where $\gamma \approx 0.5$ to include the effect of induced mass acceleration introduced by Turner (1963).

Due to significant uncertainties and complexities of entrainment and turbulent mixing (Khain et al., 2000), only lateral

entrainment that mixes in ambient air instantaneously and is homogeneous in the parcel is considered in the DCPM. Based on observations from McCarthy (1974), the entrainment rate ($\mu$) is represented by an empirical relationship that describes the




influx of air and ambient particles into the parcel as varying inversely with cloud radius. To describe the lateral entrainment, the bubble model (Scorer and Ludlam, 1953) and the jet model (Morton, 1957) are both incorporated in the parcel model.

For the bubble model, the change of the radius of a thermal bubble ($R_B$) over time is given as

$$\frac{dlnR_B}{dt} = \frac{1}{3}\left(\mu_B V - \frac{dln\rho_a}{dt}\right) \tag{4}$$

where $\mu_B = C_B/R_B$ and $C_B \approx 0.6$ (McCarthy, 1974).

For the jet model, the time variation of the radius of a jet plume ($R_J$) is expressed by

$$\frac{dlnR_J}{dt} = \frac{1}{2}\left(\mu_J V - \frac{dln\rho_a}{dt} - \frac{dlnV}{dt}\right) \tag{5}$$

where $\mu_J = C_J/R_J$ and $C_J \approx 0.2$ (Squires and Turner, 1962).

The condensational growth rate of droplets in the $i^{th}$ bin ($i = 1, 2,..., nbin$) is represented as

$$\frac{dr_i}{dt} = \frac{G}{r_i}(S - S_{eq}) \tag{6}$$

where droplet growth via condensation is driven by the difference between the ambient supersaturation ($S$) and the droplet equilibrium supersaturation ($S_{eq}$, see Eq. A4 in Appendix A). The growth coefficient ($G$) depends on the physicochemical

properties of aerosols (see Eq. A1 in Appendix A).

Assuming $S \ll 1$, then $(1+S) \approx 1$, and the time variation of the supersaturation in the parcel can be expressed as

$$\frac{dS}{dt} = \alpha V - \gamma\left(\frac{dw_L}{dt} + \mu V w_L\right) + \mu V \left[\frac{LM_w}{RT^2}(T - T') - \frac{pM_a}{e_s M_w}(w_v - w_v')\right] \tag{7}$$

where $\alpha$ and $\gamma$ depend on temperature and pressure (see Eq. A5 and A6 in Appendix A).

During the parcel's ascent, entrainment mixes out cloud droplets and interstitial aerosols inside the parcel and brings in dry air and aerosol particles from the environment. Entrained aerosols are exposed to supersaturated conditions in the parcel.

Some of them become activated and continuously grow into cloud droplets. The rate of change in droplet number in the $i^{th}$ bin ($i = 1, 2,..., nbin$) due to entrainment is

$$\left(\frac{dN_i}{dt}\right)_{ent} = -\mu V(N_i - N_i') \tag{8}$$

where the number concentration of ambient aerosol particles at a certain altitudinal level $N'(z)$ is calculated based on the assumption that the initial aerosol distribution at the surface $N(0)$ decays exponentially with height: $N'(z)=N(0)exp(-z/H_S)$, where $z$ is the height above ground level (AGL) and $H_S$ is the scale height, depending on aerosol types (Kokhanovsky and de

Leeuw, 2009).

The rate of change in liquid water mixing ratio ($w_L$) in the parcel is calculated as follows

$$\frac{dw_L}{dt} = \frac{4\pi\rho_w}{3\rho_a} \sum_{i=1}^{nbin}\left(3N_i r_i^2 \frac{dr_i}{dt} + r_i^3 \frac{dN_i}{dt}\right) \tag{9}$$



## 2.2 Collision-coalescence growth

To describe droplet growth by collision-coalescence process, the stochastic collection equation (SCE) that solves for the time rate of change in the number concentration is written following Hu and Srivastava (1995)

$$\frac{\partial N(v)}{\partial t} = \frac{1}{2}\int_0^v N(v-v',t)N(v',t)C(v-v',v')dv' - N(v,t)\int_0^\infty N(v',t)C(v,v')dv' \tag{10}$$

where the first integral on the right-hand side of the equation describes the production of droplets of volume $v$ resulting from coalescence of smaller drops, and the second integral accounts for the removal of droplets of volume $v$ due to coalescence with other droplets. The continuous SCE is discretized and numerically solved by a linear flux method as outlined by Bott (1998). This method is mass conservative, introduces minimal numerical diffusion, and is highly computationally efficient (Kerkweg et al., 2003;Pinsky and Khain, 2002). As noted before, the collision-coalescence process is calculated on a moving grid with bins modified by condensational growth at each time step.

For two colliding drops of volume of $v$ and $v'$, the coalescence kernel $C(v, v')$ in Eq. (10) is computed as the product of the gravitational collision kernel $K(v, v')$ and the coalescence efficiency $E_{coal}(v, v')$,

$$C(v,v') = K(v,v')E_{coal}(v,v') \tag{11}$$

$$K(v,v') = (9\pi/16)^{1/3}\left(v^{1/3}+v'^{1/3}\right)^2|V-V'|E_{coll}(v,v') \tag{12}$$

where $V (V')$ is the terminal velocity of drop volume $v (v')$ and $E_{coll}(v, v')$ is the corresponding collision efficiency.

The terminal velocity of cloud drops is estimated following Beard (1976) in three ranges of the particle diameter (0.5 μm–19 μm, 19 μm–1.07 mm, 1.07 mm–7 mm). Another approximation by Best (1950) is also available as an option in the model. The table of drop-drop collision efficiencies at 1-μm resolution developed by Pinsky at al. (2001) is used for $E_{coll}$. This table was created based on simulations of hydrodynamic droplet interactions over a broad range of droplet radii (1–300 μm), including collisions among small cloud droplets as well as between small cloud droplets and small raindrops. Moreover, $E_{coll}$ was derived at three pressure levels of 1,000-, 750-, and 500-mb and can be interpolated at each level of a rising cloud parcel, thus taking the increase of $E_{coll}$ with height into account. Turbulence can significantly enhance collision rates especially for small droplets (below 10 μm in radii) as it increases swept volumes and collision efficiencies, and influences the collision kernels and droplet clustering (Khain and Pinsky, 1997;Pinsky et al., 1999;Pinsky et al., 2000). Considering different turbulent intensities for typical stratiform, cumulus, and cumulonimbus clouds, detailed tables of collision kernels and efficiencies in turbulent flow, created by Pinsky et al. (2008) for cloud droplets with radii below 21 μm, are also incorporated in the model. $E_{coal}$ is parameterized following Seifert et al. (2005), who applied Beard and Ochs (1995) for small raindrops ($d_S < 300$ μm), Low and List (1982) for large raindrops ($d_S > 600$ μm), and used an interpolation formula for intermediate drops (300 μm $< d_S <$ 600 μm) where $d_S$ is the diameter of the small droplet. A simpler and faster option suggested by Beard and Ochs (1984) is also available in the model.





### 2.3 Numerical formulation

The equations in Sect. 2.1 constitute a stiff system of non-linear, first-order ordinary differential equations and involve state variables at very different scales. For the numerical integration of condensation growth, a fifth-order Runge-Kutta scheme with Cash-Karp parameters (Cash and Karp, 1990) using adaptive time steps (Press et al., 2007) is employed.

At each time step, the error is estimated using the fourth-order and the fifth-order Runge-Kutta methods. Because dependent variables differ by several orders of magnitude, a fractional error ($\varepsilon$) is defined to scale the error estimate by the magnitude of each variable. Specifically, the step size is adaptively selected to satisfy a fractional tolerance of $10^{-7}$ for all variables. The initial time step to calculate condensational growth is $5 \times 10^{-4}$ s. The maximum time step is set as $10^{-3}$ s to ensure the diffusional growth of drops is precisely simulated and non-activated particles reach equilibrium with the parcel

supersaturation at each time step. For the collision-coalescence processes in Sect. 2.2, a simple Euler method is applied to integrate forward in time, and a time increment of 0.2 s is used to reduce computation costs. Relying on separate numerical integration methods for calculating condensation and collision-coalescence allows us to either include or exclude each process easily to examine its role individually in cloud formation.

### 3 IPHEx observations

The intense observing period (IOP) of the IPHEx field campaign took place during 01 May–15 June, 2014. The study region was centred on the SAM extending to the nearby Piedmont and Coastal Plain regions of North Carolina (see maps in Fig. 1). IPHEx was one of the ground validation campaigns after the launch of NASA's Global Precipitation Mission (GPM) core satellite, and details about this campaign can be found in the science plan (Barros et al. 2014). During the IPHEx IOP, measurements of aerosol concentrations and size distributions ranging from 0.01 to 10 μm were collected at

the ground level. Collocated with aerosol instruments, the ACHIEVE (Aerosol-Cloud-Humidity Interaction Exploring & Validating Enterprise) platform was also deployed, equipped with W-band (94 GHz) and X-band (10.4 GHz) radars, a ceilometer, and a microwave radiometer. Two aircraft were dedicated to the IPHEx campaign. The NASA ER-2 carried multi-frequency radars (e.g., a dual-frequency Ka-/Ku-, W-, X-band) and radiometers, and functioned as the GPM core-satellite sampling simulator from high altitude. The University of North Dakota (UND) Citation aircraft was instrumented to

characterize the microphysics and dynamical properties of clouds, including LWC and DSDs from cloud to rainfall drop sizes. Therefore, this data set offers a great opportunity to perform modelling studies of warm season cloud formation in complex terrain. A detailed description of the specific measurements relevant to this study is provided below.

### 3.1 Surface measurements

Aerosol observations were collected at the Maggie Valley (MV) supersite (marked as the yellow star in Fig. 1b;

elevation: 925 m MSL) in the inner mountain region during the IPHEx IOP. This data set provides a clear characterization of the size distribution and hygroscopicity of surface aerosols in this inner mountain valley, which was not available previously.





Nominal dry aerosol size distributions at the surface were measured by a scanning mobility particle counter system (SMPS) for particles from 0.01 to 0.5 μm in diameter, and a passive cavity aerosol spectrometer (PCASP) for particle diameters in the size range of 0.1–10 μm. The SMPS consists of an electrostatic classifier (TSI Inc. 3081) and a condensation particle counter (CPC, TSI 3771). Note that the relative humidity (RH) of the differential mobility analyser (DMA) column is well

controlled and the average RH (± one standard deviation) of the sheath and sample flows are 2.0±0.8% and 3.2±0.5%, respectively. In addition, a co-located ambient CPC (TSI 3772), which measures aerosol particles greater than 10 nm without resolving their size distributions, shows very close agreement with the SMPS measurements with regard to total number concentrations of aerosol particles ($N_{CN}$). Size-resolved CCN concentrations ($N_{CCN}$) were sampled by a single column CCN counter (Droplet Measurement Technologies) that was operated in parallel to the SMPS-CPC. The CCN instrument cycles

through 6 levels of supersaturation (S) in the range of 0.09–0.51%. At a given S level, each CCN measurement cycle took approximately 8 mins, corresponding to one SMPS-scan and buffer time to adjust supersaturation. On average 178 measurement cycles were collected daily during the IPHEx IOP, except for occasional interruptions due to instrument maintenance. CN and CCN distributions were inverted as described previously (Nguyen et al., 2014;Petters and Petters, 2016). Supersaturation was calibrated using dried ammonium sulfate and a water activity model (Christensen and Petters,

2012;Petters and Petters, 2016). The midpoint activation diameter ($D_{50}$) is derived from the inverted CN and CCN distributions (Petters et al., 2009). The hygroscopicity parameter (κ) is obtained from $D_{50}$ and instrument supersaturation (Petters and Kreidenweis, 2007). In addition, a co-located Vaisala weather station (WXT520) was continuously recording local meteorological conditions (e.g., wind speed, wind direction, relative humidity, temperature, and pressure) at 1-s interval. Diurnal cycles of these local meteorological variables during the IPHEx IOP are displayed in Fig. S7. The average

meteorological conditions at the sampling site are 0.8±0.6 m s$^{-1}$ in wind speed, 172±115° in wind direction, 77±18% in relative humidity, 19±4 °C in ambient temperature (arithmetic mean ± one standard deviation).

Figure 3 presents a general overview of the temporal variability in aerosol size distributions and total number concentrations from the SMPS and PCASP, respectively during the entire sampling period. To avoid episodic intrusion of long-range transport or local pollution, aerosol measurements with $N_{CN,SMPS} > 10,000$ cm$^{-3}$ were removed from the analysis

in order to isolate inherent properties of aerosol particles in the pristine forest environment of the SAM. The average total number concentration (± one standard deviation) of dry aerosol particles with diameters between 0.01 to 0.5 μm is 2,487±1,239 cm$^{-3}$, as sampled by the SMPS during the campaign (see Figs. 3a and b). Strong local fluctuations in number concentrations, in particular around midnight, are due to the presence of Aitken mode particles as indicated in Fig. 3a. These sharp increases in small particles are likely produced by the power engine in the Maggie Valley Sanitary District adjacent to

the sampling site. The average total number concentration (± one standard deviation) of dry aerosol particles in accumulation and coarse modes (0.1–10 μm in diameter) is 1,106±427 cm$^{-3}$ as sampled by the PCASP during the campaign (see Figs. 3c and d). As expected, large particles from the PCASP show a much lower temporal variability in number concentrations as compared to small particles from the SMPS. Similarly, their diurnal cycles (see Figs. S8a and b) exhibit relatively large





temporal variations in $N_{CN,SMPS}$ while $N_{CN,PCASP}$ remain relatively stable throughout the day. Rainfall occurrences result in steep decreases in aerosol number concentrations, as shown in Figs. 3b and d.

As discussed before, $\kappa$ and $N_{CCN}$ were derived at six different supersaturation levels. In this study, we only show measurements collected at relatively high supersaturations (0.19–0.51%) as poor fits to $D_{50}$ are often resulted due to low

number concentration at S = 0.09% and 0.12% and thus no kappa value was reported. Figure 4 shows that both $\kappa$ and $N_{CCN}$ exhibit large temporal variabilities during the campaign. In Fig. 4a, the average value of $\kappa$ (± one standard deviation) is 0.28±0.09 at S = 0.19%, 0.22±0.08 at S = 0.38%, 0.18±0.07 at S = 0.51%. In spite of local fluctuations in $\kappa$ at each supersaturation level, larger $\kappa$ values are generally obtained at lower supersaturation (Fig. 4a). A higher value of $\kappa$ is derived from a larger $D_{50}$ due to the fact that only large particles can be activated at a low supersaturation. Therefore, aerosol

particles of different sizes are characterized with different hygroscopic properties. This is consistent with the finding from an earlier study in the Amazon rainforest showing that accumulation mode particles are more hygroscopic than Aiken mode particles (Gunthe et al., 2009). Note that the average $\kappa$ values at each supersaturation level are comparable to subsaturated $\kappa$ (0.14–0.46) measured in the southeastern United States (Nguyen et al., 2014) and the approximate global average ($\kappa \sim 0.3$) for continental aerosols (Andreae and Rosenfeld, 2008). At this surface site, the average $N_{CCN}$ (± one standard deviation) is

569±208 cm$^{-3}$ at S = 0.19%, 1,022±387 cm$^{-3}$ at S = 0.38%, 1,210±505 cm$^{-3}$ at S = 0.51% (see Fig. 4b). The diurnal cycles in Fig. S8d indicate that $N_{CCN}$ at S = 0.19% is remarkably stable while $N_{CCN}$ at higher supersaturations (0.38% and 0.51%) exhibit pronounced variations throughout the day, likely linked to the changes in small particle concentrations (see Fig. S8a). In general, no evident diurnal cycles in $\kappa$ and $N_{CCN}$ are noted from the observations in Figs. S8c and d.

**3.2 Aircraft measurements**

Airborne observations from the UND Citation, equipped with meteorological (e.g., temperature, pressure, humidity) sensors and microphysical instruments, are used in this study (Poellot, 2015). Vertical velocity was obtained from a gust probe and bulk LWC values were retrieved from two hot-wire probes (a King-type probe and a Nevzorov probe). Size-resolved concentrations were measured from three optical probes, covering droplet diameter from 50 μm to 3 cm: a PMS two-dimensional cloud probe (2D-C), a SPEC two-dimensional stereo probe (2D-S), and a SPEC high volume precipitation

spectrometer 3 probe (HVPS-3). The cloud droplet probe (CDP) measures cloud drop concentrations and size distributions for small particles with diameters from 2 to 50 μm in 30 bin sizes. The droplet sizes are determined by measuring the forward scattering intensity when droplets transit the sample area of the CDP. Coincidence errors have been found to cause measurement artefacts, which tend to underestimate droplet concentrations and broaden droplet spectra. This type of error occurs when two or more droplets pass through the CDP laser beam simultaneously, and is highly dependent on droplet

concentrations (Lance et al., 2010).

Bulk LWC measurements from hot-wire probes can serve as independent observations to identify and correct coincidence-related sizing errors in the CDP. For example, during the flight on 12 June 2014, bulk LWC values from the King and Nevzorov probes are used to evaluate the CDP-derived LWC integrated from its droplet size distribution (see Eq.





A7 in Appendix A). In this study, bulk LWC data with air temperature greater than 0 °C are considered in order to eliminate erroneous attribution of ice- or mix-phase particles to liquid water by hot-wire probes. In Fig. 5a, we can notice that CDP LWC produces a positive bias compared to LWC from the two hot-wire probes, whereas the King and Nevzorov probes demonstrate general agreement with each other. The CDP instrument aboard the UND Citation was modified by adding an

optical mask, which has been proven to resolve the underestimation of droplet concentrations (Delene, 2016;Lance, 2012). Herein, we assume that the bias in CDP LWC is caused by the oversizing error rather than the undercounting error. Thus, we applied a correction to the CDP size distributions, as introduced by Painemal and Zuidema (2011). This bias can be removed based on the linear correlation revealed by the comparison between the King- and CDP-LWC using data collected during the first horizontal leg of the 12 June flight (see Fig. 5b). In the correction procedure, King LWC data between 0.05 and 0.6 g m$^{-3}$

are taken into account. Thus, a linear regression with coefficient of determination $R^2 = 0.80$ is fitted between the CDP- and King-LWC and the derived slope (= 1.26 as denoted in Fig. 5b) is used to adjust CDP droplet size distributions. The modified droplet size in each bin is calculated by dividing the original size by $1.26^{1/3}$ (~ 1.08) to attain consistent LWC between the CDP and the King probe. The corrected droplet size distributions slightly shift the measured spectra to smaller drop sizes (not shown here), thus providing confidence in the performance of the CDP probe during the campaign.

### 3.3 Cumulus congestus case-study

On 12 June 2014, cumulus congestus clouds were observed by the W-band radar (see Fig. 6) at MV, and they were also sampled by a coordinated aircraft mission of both the UND Citation and NASR ER-2 flying near the MV site. The flight period of the UND Citation on 12 June is from 12:14 to 15:51 local time (LT). Cloud droplet concentrations and size distributions were sampled by conducting successively higher constant-altitude flight transects through clouds. Droplet

spectra were sampled at 1-Hz resolution (corresponding to approximately 90 m in flight distance) by the CDP and coincidence errors were taken into account by applying the correction as described in Sect. 3.2. In particular, the lowest horizontal leg (see the flight track in Fig. 7a, altitude around 2,770–2,800 m MSL) through the cloud is investigated to avoid the influence of substantial mixing in the upper portion of the cloud, which is not treated in the DCPM currently. In rising updrafts, in-cloud samples (white plus signs in Fig. 7a) are defined with a minimum LWC of 0.25 g m$^{-3}$ from the CDP. To

further eliminate regions influenced by mixing and other unresolved mechanisms, cloud segments to perform the modelling study are carefully selected by screening the cloud droplet spectra observed by the CDP. Following criteria 2 and 3 listed in Conant et al. (2004), measurements with effective droplet diameter greater than 2.4 μm and geometric standard deviation less than 1.5 are used in the analysis. During the first cloud transect, only one in-cloud region (IC, circled in Fig. 7a) satisfies Conant et al.'s requirements with 11 cloudy samples collected over approximately 1 km flight distance (plus signs in Fig.

7b). Significant topographic heterogeneity (see terrain transect in Fig. 7b) can exert a considerable influence on cloud formation across this region. As shown in Figs. 7c and d, a pronounced variability in drop number distributions is manifest in the in-cloud samples clustered by low (0–1 m s$^{-1}$) and high (1–2 m s$^{-1}$) updrafts. As expected, the droplet spectra in stronger updrafts at the core (see Fig. 7d) have higher number concentrations and narrower size range compared to the samples at the



edge of the cloud (see Fig. 7c). Observed variations in vertical velocities and droplet number concentrations in complex terrain are indicative of challenges in the application of parcel models as homogeneity is assumed for aerosol concentrations below cloud base and within the microstructure of the air parcel.

Moreover, droplet spectra measured within updraft core of two other cloudy regions in the inner SAM (highlighted in dashed light blue boxes in Fig. 7a) as well IC are shown in the top panel of Fig. 8. Fig. 8d displays the background aerosol concentrations measured by the CPC (lower cut-off diameter 10 nm) aboard the UND Citation along the complex terrain of the SAM (elevation along the flight transect is indicated by the black line) during the first horizontal leg (see flight track in Fig. 7a). From east to west (flight direction as indicated by the blue arrow), it is noticeable that the three cloud regions (shaded in Fig. 8d) are linked to considerable drops in the aerosol concentrations. In particular, clouds form over the foothills of the eastern ridges (ER, see location in Fig. 7a) in the inner region are associated with low-level moisture convergence from the east (Wilson and Barros, 2017). The cloud core sampled in this convergence zone is formed in intense updrafts (~ 8 m s$^{-1}$, see Fig. 8c) and it exhibits wide droplet spectra with heavier tails (larger drops) than the observations in the IC core (updrafts ~ 1–2 m s$^{-1}$, see Fig. 8b). The in-cloud samples over high terrain elevations near the Eastern Cherokee Reservation (ECR, see location in Fig. 7a) also exhibit broad spectra but smaller number concentrations suggesting strong mixing with its environments, likely related to mid-day ridge-valley circulations (Wilson and Barros, 2017). As noted in Fig. 8d, significant increases (~ 1,000 cm$^{-3}$) in aerosol number concentrations are evident when the aircraft flew from the French Board (FB) valley into the inner SAM region that includes the Pisgah National Forest and the Great Smoky Mountains National Park (Figs. 1b and 7a). Generally, there is a close agreement between salient topographic features and variations in aerosol number concentrations. As size distributions are not resolved in the CPC measurements, we resort to the surface sampling of aerosol concentrations at MV (indicated by the dashed vertical line in Fig. 8d and marked as the black asterisk in Fig. 7a) as the input for modelling study at IC. Moderate vertical velocities measured in IC region (Fig. 7b) and analysis of the radar profiles at MV (Fig. 6) suggest that the early development phase of the cumulus congestus observed in the inner SAM was sampled by the aircraft on June 12 during IPHEx.

## 4 Modelling experiments

### 4.1 Model initialization and reference simulation

Dry aerosol concentrations measured by the SMPS and PCASP at MV were averaged over the first 10 mins (averaging interval: 12:14 LT–12:24 LT) of the 12 June flight, and then merged into a single size distribution as shown in Fig. 9. The combined aerosol distribution at the surface is fitted by the superimposition of four lognormal functions using least-squares minimization. Table 2 summarizes parameters (total number concentration, geometric mean diameter, and geometric standard deviation) that characterize the four lognormal distributions. We can notice that aerosol number concentrations below 0.03 μm are greatly underestimated by the fitted cumulative distribution (cyan curve in Fig. 9). These particles in such small sizes mostly remain non-activated under the supersaturated condition typical of the atmosphere, thus,





underestimation of their concentrations does not significantly affect our modelling results of cloud formation. At the cloud base of IC, aerosol size distributions are estimated by assuming that total number concentrations at the surface decay exponentially with a scale height ($H_S$) of 1,000 m, and geometric mean diameters and corresponding geometric standard deviations remain constant with height. The dry aerosol distribution at cloud base is calculated as the sum of four lognormal

distributions with fitted parameters indicated in the last three columns of Table 2 and is taken as initial input to the model. The aerosol distribution is discretized into 1,000 bins, covering the size range of 0.01–10 μm. The bins are spaced geometrically with a volume ratio of 1.026. The bin grid at such a high resolution is sufficient to precisely simulate the partitioning of growing droplets and interstitial aerosols in the parcel. It is also assumed that the aerosol is internally mixed so that the hygroscopicity does not vary with particle size. Thus, we prescribe a κ value of 0.14 for each aerosol bin, deriving

from the average κ during the first 10 mins of the 12 June flight.

During the IPHEx IOP, daytime radiosondes were launched every 3-hour at Asheville, NC (see the red star in Fig. 1b). Not only is this location very far away from the targeted cloudy region (IC), but the timing of the sounding (11 LT) is also much earlier than the flight take-off on 12 June 2014. At 11 LT, the sounding at Asheville shows a relatively dry atmosphere especially at low levels (not shown here). In this study, we resort to high-resolution WRF model simulations to

provide vertical profiles of air temperature, RH, and pressure as sounding input to the parcel model. Detailed domain configuration of the WRF simulation (see Fig. 10a) can be found in Sect. S2. WRF-simulated sounding columns from the grid cells (0.25-km resolution) in the IC region (highlighted in Fig. 7a) are averaged to estimate vertical profiles of ambient temperature and RH for the case-study as shown in Fig. 10b. The cloud base height (CBH) is chosen as the level where simulated RH is approximately 100%. As marked by the horizontal black line in Fig. 10b, CBH = 1,270 m AGL at 12:15 LT

when the parcel is released from cloud base. The temperature excess of the air parcel over the environment is initialized as 1.0 K, and the initial pressure and RH of the parcel at cloud base adapt to cloud surroundings. As vertical velocities were not sampled at cloud base, the initial updraft velocity ($V_0$) is assumed to be uniformly distributed and equal to 0.5 m s$^{-1}$, consistent with vertical velocities observed by the W-band radar (see Fig. 6b) around the same altitude (2.5 km MSL). Therefore, the air parcel is launched at cloud base with an initial parcel radius (R) of 500 m, an initial updraft of 0.5 m s$^{-1}$,

and initial aerosol particles that are in equilibrium with the humid air at cloud base. When the parcel is rising, the lateral entrainment is treated as the bubble model parameterization with the characteristic length scale R = 500 m (see Eq. 4 in Sect. 2.1). Ambient aerosol particles penetrate through lateral parcel boundaries and their number concentrations also decrease exponentially with height ($H_S$ = 1,000 m). The turbulent kinetic energy dissipation rate is chosen as 200 cm$^2$s$^{-3}$, typical of cumulus clouds at early stages. The parcel reaches cloud top when vertical velocity is near zero. Note that despite specified

as stated above for the reference simulation, sensitivity to parcel radius R and scale height Hs will also be explored in Sect. 4.2.



### 4.2 Parameter sensitivity analysis

In this section, sensitivity tests are conducted to probe the range of unavailable measurements in light of in-cloud observations from the aircraft and assess the role of individual state variables and processes on the microphysics of the cumulus congestus case-study on 12 June during IPHEx. Selected parameters are perturbed one at a time while other
assumptions and input parameters remain as specified in Sect. 4.1.

### 4.2.1 Condensation coefficient

Condensation plays a dominant role in the early stages of cloud formation and one key factor in this process is the condensation coefficient ($a_c$) that governs activation and condensational growth. A laboratory study by Chuang (2003) reported $a_c$ values ranging from $4 \times 10^{-5}$–1, and experimental values from field campaigns and from chamber studies of
individual droplet growth also differ over a wide range (0.007–0.1) as reviewed in Sect. 1. To determine an optimal value of the condensation coefficient that achieves a close agreement with the IPHEx airborne observations, $a_c$ was made to vary in the range [0.001–1.0] on the basis of Fountoukis and Nenes (2005). For the targeted in-cloud region (IC), Fig. 11 shows simulated profiles of updraft velocity, supersaturation, total CDNC, LWC, and their sensitivity to selected $a_c$ values in comparison with the airborne observations (different symbols indicate the ranges of measured updraft velocities triangles: 0–
0.5 m s$^{-1}$, squares: 0.5–1.0 m s$^{-1}$, pentagrams: 1–1.5 m s$^{-1}$, hexagrams: 1.5–2.0 m s$^{-1}$). Measurements from the IC region along the lowest cloud transect (highlighted in the blue circle in Fig. 7a) are used to evaluate model performance, since no observations are available in the upper unmixed cloudy areas to assess the entire vertical profiles simulated by the CPM. Only simulations with reasonable agreement with the observations are presented, thus results with $a_c$ from 0.06 to 1.0 are not shown here. Particles above 1 μm in diameter are considered cloud droplets and are included in the integration to
calculate LWC. Note that ground elevations under the IC region vary from 928 m to 1,184 m MSL (see Fig. 7b), and the region is on a small hill in the middle of the valley and surrounded by much higher ridges (terrain elevation ~ 1,500 m MSL). Hereafter, aircraft measurements are expressed as AGL to facilitate their comparisons with the model results.

Simulated updraft velocities at the observation levels (Fig. 11a) are consistent with the general trend of airborne measurements, which decrease with height. It is apparent that $a_c$ has a significant impact on the simulated supersaturation
profiles (Fig. 11b). Low values of $a_c$ strongly inhibit the phase transfer of water vapour molecules onto aerosol particles, slowing the depletion of water vapour available in the parcel, and thus substantially increasing maximum supersaturation ($S_{max}$). Consequently, smaller aerosol particles with high concentrations are activated due to a higher $S_{max}$ further up from the cloud base, resulting in a direct increase in cloud droplet numbers (Fig. 11c). Overall, the results are in agreement with earlier studies (Nenes et al., 2002;Simmel et al., 2005) that investigated the dependence of cloud droplet number
concentrations on the condensation coefficient. Moreover, Fig. 11c shows that the simulation with $a_c = 0.01$ (green line) captures well the observed drop concentrations between 1,500 m and 1,600 m AGL (highlighted in yellow shade), whereas a condensation coefficient that is one order of magnitude lower (blue line) yields better results for the observations above





1,600 m. As summarized in Table 3, the average CDNC simulated for the region between 1,500 m and 1,600 m AGL along the hillslope (shaded in Fig. 7b, reference sub-region within IC) for $a_c$ = 0.01 attains an average CDNC of 354 cm$^{-3}$, which is only ~1.3% higher than the observed average between 1,500 m and 1,600 m (349.4 cm$^{-3}$). The corresponding LWC is also in reasonable agreement with the range of observed values (Fig. 11d). The closure error for the upper cluster between 1,600 m

and 1,750 m using a much lower condensation coefficient (0.002) is ~ 7% (blue line). The simulated supersaturations indicate strong sensitivity to $a_c$, however, all underestimate the measured values significantly, which is attributed to the sounding input obtained the WRF simulation that appears to be drier and warmer than the local atmospheric conditions. This issue will be discussed again below and in Appendix B1 to assess the impact of adjusting the sounding on cloud growth. Inspection of Fig. 11c suggests that within IC there are two clusters of air parcels at different levels above ground associated

with different condensation coefficients. Interestingly, the higher cluster (above lower elevation, Fig. 7b) is better matched by a lower condensation coefficient, whereas a higher condensation coefficient leads to better closure in the reference region that includes the updraft core near the hilltop.

The sensitivity of predicted spectra at 1,500 m (in solid lines, Fig. 12a) to $a_c$ varying from 0.002 to 0.06 is very high. The observed spectrum (black dotted line) is the average from five individual CDP measurements (dotted lines with

circle markers in Figs. 7c and d, also highlighted in the yellow shaded area in Fig. 7b) between 1,500 m and 1,600 m AGL (see Fig. 11d for their LWC in shade). Generally, spectra simulated with lower values of $a_c$ are broader with higher numbers of small droplets, while simulations with large values of $a_c$ yield narrower spectra shifted to larger droplet sizes. The differences in drop size range and spectra shape can be explained by inspecting the vertical profiles of the parcel supersaturation and $S_{eq}$ for six illustrative aerosol particle diameter ($D_{aero}$) depicted in Fig. S9. Growth by water vapour

condensing on different sizes of cloud droplets is determined by the difference between S and $S_{eq}$ (Eq. 6 in Sect. 2.1). At low S, small particles become interstitial aerosols, and their corresponding $S_{eq}$ remains in equilibrium with the parcel supersaturation (S - $S_{eq}$ = 0). At high S, as a result of low $a_c$, activation of small aerosols contributes to significant spectra broadening, produces larger CDNC, and shifts the droplet size distribution toward smaller diameters due to slower condensational growth. This is consistent with Warner (1969a) who found that low condensation coefficients (< 0.05) were

required to capture the observed dispersion of droplet spectra in natural clouds, especially for small sizes (i.e. left-hand side of the spectra). Figure 12b displays the simulated droplet number distributions at different levels for $a_c$ = 0.01 in comparison with the individual droplet spectra measured by the CDP. The spectrum at 1,559 m AGL (black dotted line) and its CDNC (357 cm$^{-3}$) and LWC (0.37 g m$^{-3}$) are closely replicated by the DCPM, and the simulated spectra are representative of the evolution of cloud droplet distributions in one parcel at different cloud development stages. Simulated spectra at 1,500 m and

1,600 m altitude show very good agreement with the observed number concentration and drop size range. Below 1,600 m, a shift of the unimodal spectra to larger drop sizes suggests that the condensation process currently dominates the growth of cloud droplets. Larger drops above 1,700 m could be produced by coalescence growth, leading to the formation of a second mode at larger sizes in the upper portion of the cloud. For the analyses presented hereafter, we consider $a_c$ = 0.01 together





with other initial conditions as prescribed in Sect. 4.1, as the reference simulation (denoted by the grey line in the following figures).

Further examination using data from other cloud and precipitation probes suggests that concentrations of droplets larger than 30 μm in diameter are negligible during the first horizontal leg. Considering that droplets with diameters larger

than 30–32 μm are required to trigger effective droplet collisions (Pinsky and Khain, 2002), we conclude that the collision-coalescence process is not important in the sampled IC region, and it is unlikely that it contributes to the wide bimodal spectra observed at early stages of cloud growth. It is noteworthy that small drops are absent in the simulated spectra, in contrast to the observed spectrum that exhibits a broad drop size range and two distinct modes (see Fig. 12b). One possible explanation is that the moving bin grid determined by the condensation process tends to widen the spectral gap between the

growing droplets and non-activated aerosol particles in the ascending parcel. Thus, a geometric size distribution with 1,000 bins is utilized herein to further refine the discretization for small particle sizes. Another explanation relates to the uncertainties of the input sounding extracted from the WRF simulation. Even though ambient aerosols are continuously entrained through lateral boundaries, most of them remain as interstitial aerosol particles because the low supersaturation in the parcel is insufficient to enable activation (see Fig. 11b). The WRF sounding in Fig. 10b exhibits a lapse rate of -4.1 °C

km$^{-1}$ from 1,270 m (CBH) to 2,200 m, corresponding to stable atmospheric conditions unfavourable for cloud development. Additional model simulations were performed by altering lapse rates and humidity profiles at lower levels (see Appendix B1). The results point out that uncertainties of the assumed environmental thermodynamic conditions (e.g., temperature, humidity) impose significant constraints in reproducing wider bimodal spectra present in natural clouds, thus posing as a significant challenge in cloud modelling study.

**4.2.2 Entrainment strength**

To access the influence of entrainment on cloud drop concentrations and LWC, different strengths of lateral entrainment are examined by altering the initial cloud parcel size R at the cloud base. Figure 13 displays the vertical profiles of total CDNC and LWC, and cloud droplet spectra formed at three altitudinal levels (1,500 m: solid line, 1,600 m: dotted line, and 1,700 m: dashed line) for simulations using different initial parcel radii as compared to the CDP observations in the

IC region (denoted by black symbols in Figs. 13a and b and the black dotted line in Fig. 13c). Entrainment appears to have a dominant influence on the cloud vertical structure as small rising parcels associated with higher entrainment dissipate faster by intensive mixing of dry ambient air through lateral cloud boundaries. Stronger entrainment strength results in a direct decrease in drop concentrations and LWC, while it has little influence on the droplet size range. The best agreement on droplet numbers is between the reference simulation (R = 500 m, $a_c$ = 0.01; grey line in Fig. 13a) and the reference sub-

region within IC (between 1,500 m and 1,600 m AGL), whereas results for R = 1,500 m better replicate the higher cluster of cloudy samples (above 1,600 m AGL). Recall that previously, when R was held constant the higher cluster is better reproduced using $a_c$ values one order of magnitude smaller than the reference value. Thus, the sensitivity analysis does





suggest there is a trade-off with weaker entrainment for a higher condensation coefficient (R = 1500 m and $a_c$ = 0.01, the orange line in Fig. 13a) when other parameters in the reference simulation remain the same.

Given R = 500 m, an additional test was conducted using the jet model parametrization of lateral entrainment (Eq. 5 in Sect. 2.1). The comparison of two entrainment parameterizations indicates that the bubble model (the grey line) has
stronger entrainment strength than the jet model (red line) given the same initial parcel size (500 m). Nevertheless, continuous increases in simulated LWC in the upper portion of the cloud (see Fig. 13b) for both parameterizations are unrealistic in real clouds (Paluch, 1979). This problem can be likely ascribed, at least in part, to the uncertainties in the environmental conditions associated with the WRF sounding. As noted in Fig. B1, decreases in LWC are manifest at the upper portion of the cloud, as indicated in the simulations with modified sounding inputs. The lack of sufficient mixing with
dry ambient air near cloud top is an inherent deficiency in the simple parameterization of lateral homogenous entrainment, assuming decreasing entrainment strength with height, but this assumption does not significantly affect our conclusions for in-cloud regions below cloud top.

### 4.2.3 Initial aerosol concentration

The initial aerosol concentration at cloud base can also have significant effects on cloud development. Because
aerosol size distributions were not sampled by the aircraft during IPHEx, they are estimated by extrapolating surface aerosol number concentrations according to an exponential decay with a given scale height ($H_S$). To probe and characterize the dependence of droplet formation on aerosol concentrations available at cloud base, sensitivity to $H_S$ was explored by varying its values from 800 m to 1,200 m. Figure 14 shows the simulated profiles of the total CDNC and LWC, and cloud droplet spectra formed at three altitudinal levels (1,500 m: solid line, 1,600 m: dotted line, and 1,700 m: dashed line). It is not
surprising that aerosol concentrations at cloud base have a substantial influence on the resulting droplet concentrations. Higher aerosol concentrations, inferred from larger $H_S$ lead to larger drop numbers with smaller average droplet sizes, which is known as the first indirect effect of aerosols (Twomey, 1977). Yet, here, LWC appears insensitive to the initial aerosol concentration as it is limited by moisture content available in the parcel. The optimal value of $H_S$ = 1,000 m yields the best agreement on CDNC between the DCPM simulations and the average droplet spectra observed by the CDP (black dotted line
in Fig. 14c, see reference sub-region within IC shaded in Fig. 7b), which lies within the typical $H_S$ range (550–1,100 m) of aerosol number concentration measurements for remote continental type (Jaenicke, 1993). As noted before, aerosols in the atmosphere exhibit a significant space-time variability especially in regions of complex terrain, which can contribute to the diverse cloud droplet spectra observed across the cloud transect (see Fig. 8). We should recognize that this feature cannot be captured by current model simulations that assume a homogenous aerosol distribution at cloud base.





### 4.2.4 Summary of sensitivity analysis

Sensitivity tests by changing $a_c$ in the range of 0.001–1.0 suggest that the predicted CDNC, LWC, and thermodynamic conditions are highly dependent on the condensation coefficient with $a_c = 0.01$ achieving the best agreement with the total number concentration and size distributions from the airborne observations. At early stages of cloud

development, the condensation coefficient plays a key role in the simulated spectra width and shape that increases in $a_c$ lead to a shift towards larger droplet sizes and narrower spectra widths. Entrainment has a substantial impact on the cloud depth, droplet numbers, and LWC, whereas initial aerosol concentrations have a strong effect on number concentrations and size distributions of cloud droplets, but induce little effects on LWC. Additional tests regarding the hygroscopicity and initial updraft velocity were conducted and discussed in Appendix B2 and B3, respectively.

In this study, the cloud spectra observed in the inner region of the SAM for early development of cumulus congestus on 12 June are better reproduced by a relatively low value of $a_c$ (0.01). Previous field campaigns, as discussed in Sect. 1, have applied $a_c = 0.06$ for wam cumulus during CRYSTAL-FACE (Conant et al., 2004), $a_c = 0.042$ for stratocumulus during Coastal Stratocumulus Imposed Pertubation Experiment (CSTRIPE, Meskhidze et al., 2005), and $a_c = 0.06$ for cumuliform and stratiform clouds during ICARTT (Fountoukis et al., 2007), which are typical values for aged

atmospheric droplets (Fountoukis and Nenes, 2005;Shaw and Lamb, 1999). It is noteworthy that the aforementioned closure studies assume adiabatic conditions in the parcel and some were only evaluated near cloud base. In our study, the vertical strucuture of cloud microphysics is simulated by the DCPM with entrainment included and comparisons against the observations are performed several hundred meters above cloud base. Exploratory simulations assuming a higher aerosol number concentration at cloud base ($H_S = 1,200$ m, Fig. B4b) show a highly nonlinear response to changes in $a_c$ and R that

the aerosol-CDNC closure is achieved with higher $a_c$ values (0.03 and 0.06) for weak entrainment (R = 1500 m) consistent with the trade-off between entrainment (stronger) and condensation coefficient (lower) discussed in Sect. 4.2.2. Further, this alternative closure agreement, however, results in large discrepancies between the width and shape of simulated and observed spectra within IC (not shown here), and thus predictions of inferior skill regarding vertical development of clouds. That is, aerosol-CDNC closure near cloud base is an insufficient metric of ACI processes.

Besides entrainment, one possible explanation for the lower condensation coefficients in IPHEx can be the presence of organic film-forming compounds (FFCs) on the surface of natural aerosol particles (Feingold and Chuang, 2002). Organic films can strongly impede the uptake of moisture by atmospheric aerosol particles, thus reducing the value of $a_c$ (Gill et al., 1983;Mozurkewich, 1986). Nenes et al. (2002) conducted a parcel model study to investigate the impact of aerosol coating with organic FFCs using a constant $a_c$ and concluded that the initial condensational growth is impeded, leading to higher

supersaturations in the parcel, and increasing the cloud droplet number by a substantial amount due to a higher number of activated CCN. Without the characterization of the organic speciation in this campaign, the presence of organic coating on local aerosols remains an open question. Nevertheless, results from laboratory experiments of direct contact condensation on aerosols in cloud chambers with horizontal or vertical moist flows, point to $a_c$ values around 0.01 (Garnier et al.,



1987;Hagen et al., 1989) in contrast with the most probable value (0.06) found in the levitation cell by Shaw and Lamb (1999).

### 4.3 Multi-parcel simulations

In the previous section, it is demonstrated that single parcel simulations provide reasonable first estimates of key
parameters in cloud formation, and their results agree well with the observed ranges of in-cloud measurements during IPHEx. The formation of cumulus clouds proper can be further described as the ensemble of multiple individual rising thermals of different sizes and vertical velocities (Cooper, 1989;Warner, 1969b). In order to simulate the realistic evolution of cumulus clouds with multiple air parcels rising in succession, Mason and Jonas (1974) and Roesner et al. (1990) pursued a multi-parcel modelling strategy such that a new parcel ascends within an environment established by its predecessors.
However, when a series of parcels is rising, complicated interactions may occur under the Lagrangian framework of the parcel model, and they are difficult to resolve unless a more complex formulation of in-cloud parcel interactions is implemented (Khain et al., 2000). To examine the impact of antecedent conditions on individual cloud parcels, a simple solution is to impose a time delay for the launch of the next parcel to prevent it from overtaking the previous parcel during its ascent. As illustrated in Fig. 15, the second parcel rises through the environment modified by the first parcel ($\Gamma_1$) and
entrains interstitial aerosols left behind by the first parcel. After the second parcel ascends above the maximum height of the first parcel, it continues to rise through the undisturbed environment ($\Gamma_{env}$) and entrains ambient aerosol particles.

For the cumulus congestus clouds on 12 June 2014, a time-delay was adopted in the multi-parcel simulations so that the current parcel only passes its predecessor when it comes to rest. Results from the multi-parcel simulations show that the second parcel attains a slightly higher maximum supersaturation (1.17%) than the previous parcel (1.16%), and rises with a
higher updraft velocity (not shown here). After the second parcel emerges from the environment modified by the first parcel, no further increases in the supersaturation were produced in our simulation (not shown here), which is different from the results in Roesner et al. (1990). This is attributed to the initial environmental conditions obtained from the WRF sounding for this case-study (absolutely stable atmosphere with slow cooling below 2,200 m). Vertical profiles of drop concentrations and LWC, and droplet spectra of the first and second parcels are displayed in Fig. 16. Within the maximum height reached
by the first parcel, the second parcel experiences a steeper decrease of droplet number concentration compared to the first one because only interstitial aerosol particles are entrained and most of them remain non-activated due to only slightly increased maximum supersaturation in the second parcel. In the new environment established by the first parcel, the second parcel achieves a higher LWC value and its cloud droplet spectra (represented by the solid lines in Fig. 16c) is slightly broader extending to larger size ranges compared to the first parcel (represented by the dotted lines in Fig. 16c). Above the
maximum height of the first parcel, a pronounced secondary mode develops in the second parcel, resulting from coalescence growth tied to faster condensational growth. In addition, a third parcel simulation was conducted and its results only show slightly changes as compared to the second parcel, again likely explained by the unfavourable environmental conditions from the WRF sounding. Nevertheless, each successive air parcel can create a new thermodynamic condition for the




subsequent parcels and lead to deeper vertical development and faster droplet growth, thus conducive to convective cloud formation.

## 5 Summary and discussion

In this study, an entraining cloud parcel model (DCPM) with explicit bin microphyscis is used to explore the
vertical structure of cloud development and evaluated against extensive data collected during the IPHEx campaign during May–June 2014 in the complex terrain of the SAM (Barros et al. 2014). Because measurements of key input parameters are not available from the campaign, or cannot be resolved by current sampling techniques, there is a pressing need to investigate the physical space of such parameters (e.g., condensation coefficient, entrainment, and scale height) and their interdependencies, which ultimately govern ACI in cloud formation and development. The study specifically focuses on the
development of a mid-day cumulus congestus case on 12 June 2014 when aircraft measurements are available. Although this flight sampled three distinct cloud regions during the lowest cloud transect, the IC region and the MV supersite are closely located in the inner valley region of the SAM and thus a detailed modelling study could be conducted leveraging ground based aerosol measurements, ceilometer, and multi-frequency radar profiles available at MV to inform model initialization (Figs. 6 and 9). Given the specific set of initial conditions inferred from MV observations and initial parameters from the
literature, sensitivity analysis was first conducted to determine the possible ranges of key ACI modelling parameters that minimized aerosol-CDNC closure error for the samples collected roughly 300–500 m above cloud base in the cloud updraft core. Albeit a large variability in cloud microphysical properties was observed at sub-km scale (~ 90 m is the spatial resolution of the measurements along the flight track) over the complex terrain of the inner SAM even within IC, the modelling results for the reference simulation demonstrate good agreement with the measured LWC and droplet size spectra
of the cumulus congestus cloud.

In the framework of the physically based cloud parcel model, sensitivity of the simulated cloud microphysical characteristics to variations in key parameters was investigated within the context of in situ measurements. Results from sensitivity tests show that condensation coefficient exerts a profound influence on the droplet concentration, size distribution, LWC, and thermodynamic conditions inside the parcel, with a decrease in $a_c$ leading to an increase in cloud
droplet number, a broader droplet spectra, and a higher maximum supersaturation further up from cloud base. The case-study during IPHEx reveals that the observed cloud features in the inner mountain region of the SAM are better reproduced by a low value of $a_c$ (0.01), achieving aerosol-CDNC closure to ~1.3% of the observation. As expected, entrainment is found to be a major process controlling the vertical structure, CDNC, and LWC of the cloud. Further, it was shown that with other input parameters remain the same as reference simulation conditions, there is a trade-off between entrainment and the
condensation coefficient: strong entrainment (meaning the characteristic scale R in the bubble parameterization is small) is compensated by lower $a_c$ values, and vice-versa. This explains higher values found in previous aerosol-CDNC closure studies assuming adiabatic cloud conditions (zero entrainment) in the CPMs. Initial aerosol concentrations at cloud base also



have a large impact on droplet numbers but negligible influence on LWC. Nevertheless, analysis of the effect of the interdependence of initial aerosol concentration, condensation coefficient and entrainment strength on the CDNC revealed ambiguous behavior that could only be resolved by assessing the properties of the simulated droplet spectra (shape, range) against the aircraft measurements at different altitudes throughout the clouds (i.e., well above cloud base). Overall, these

findings provide a better picture of dominant factors in modelling cloud formation and provide some insight into key parameters of ACI processes in this region. This further highlights the need to have a constraining set of observational inputs in order to validate our findings over the SAM.

Finally, model and data limitations should be acknowledged. First, realistic entrainment and mixing with cloud surroundings have been found to contribute significantly to droplet spectrum broadening. It is important to recognize the

limitations of the lateral homogeneous entrainment employed in the model. Its concept is based on a simple assumption that entrained aerosols are mixed instantly across the parcel, which neglects the inhomogeneous supersaturation and microphysical structure inside the cloud associated with discrete entrainment events on different spatial scales (Baker et al., 1980;Khain et al., 2000). Turbulent mixing (Krueger et al., 1997) can break down entrained blobs of air into smaller scales and subsequently form small adjacent regions with uniform properties on account of molecular diffusion, thus leading to

considerable spectrum broadening. In addition, the parameterization of entrainment through lateral boundaries neglects entrainment with dry air at cloud top that is expected to be an important element to cloud vertical development (Telford et al., 1984). Downdrafts induced by the penetration of dry air at cloud top can sink and mix with updrafts, effectively diluting number concentrations and broadening droplet spectra in clouds (Telford and Chai, 1980).

The multi-parcel approach was adopted to explore the impact of thermodynamic conditions on cloud vertical

development, and consequently cloud microphysical structure. The new environment created by its predecessors enables the following air parcel to reach a higher altitude and develop larger droplets, thus facilitating the formation of convective clouds even under unfavourable environmental conditions (i.e., WRF sounding). When atmospheric soundings representative of local conditions are available, one could envision multiple parcels being lifted from cloud base at different times with different velocities over a duration sufficient to grow cloud droplets to the observed sizes. For cloud layers with thermal

instability, complexity of in-cloud vertical velocity fields with localized areas of much stronger updrafts has been found to support the formation of wide bimodal spectra in cumulus clouds due to in-cloud nucleation of new droplets from interstitial aerosols when the parcel supersaturation higher up in the cloud exceeds the cloud base maximum (Pinsky and Khain, 2002). As a result, this mechanism can lead to the formation of a secondary mode of small droplets in individual spectra, different from our observed spectra with a second mode centred at a larger droplet size (Figs. 7 and 12). High supersaturations in the

range of 1.7–3.2% are indeed measured in the lower portion of the cloud in the IC region (marked as black symbols in Fig. 11b). However, lower supersaturation is predicted by the parcel model at the observation levels and no further rise of supersaturation is present above the cloud base maximum under the conditions of the original and modified environments, likely attributed to the ambiguities in the sounding input from WRF. Therefore, the uncertainties of ambient thermodynamic conditions significantly constrain the modelling study of the observed clouds in our case. Another limitation in the current





approach is the assumption of uniform hygroscopic properties for all particle sizes. In reality, the aerosol distribution is an aggregate of particles with different physicochemical properties, including different shapes, solubility, and chemical species (Kreidenweis et al., 2003;Nenes et al., 2002). Even if specified initial aerosol characteristics were to capture the variation of κ with size, how to track the evolution of κ as particles among different bins undergo coalescence and breakup remains a

challenge. Nevertheless, the sensitivity analysis indicates that the cloud droplet growth is generally insensitive to hygroscopicity (Appendix B2), thus the constant κ value used in this study does not significantly affect our modelling results.

The present study underlines the importance of the relationhsip between entrainment processes that determine the local- (microscale) and cloud-scale thermodynamic environment around individual particles, and the aerosol condensation

coefficient that measures the effectiveness of condensation processes in the same thermodynamic environment. Given the multiscale thermodynamic structure of clouds,  these interactions suggest that realistically the condensation coefficients in the natural environment are transient and spatially variable. Therefore, further research to arrive at representative ensemble estimates are necessary to reduce the associated uncertainties of the aerosol indirect effect. In the present study, the local sensitivity of selected model parameters are assessed individually over certain ranges based on IPHEx data and the literature,

which ignores non-linear interactions among ACI modelling parameters as discussed above. Future work will focus on exploring the sensitivity of the DCPM in a multi-dimentional parameter space to quantify multiple parameter interactions (Gebremichael and Barros, 2006;Yildiz and Barros, 2007) on ACI processes using the fractorial design method (Box et al., 1978).

**Appendix A**

*Glossary of Symbols*

$a_c$     condensation coefficient

$a_T$     thermal accommodation coefficient

$c_p$     specific heat of dry air

$D_v, D_v'$   diffusivity of water vapor in air, modified diffusivity of water vapor in air

$e_s$     saturation vapor pressure

$g$     gravitational constant

$G$     growth coefficient

$H_S$     scale height

$k_a, k_a'$   thermal conductivity of air, modified thermal conductivity of air

$L$     latent heat of evaporation

$M_a, M_w$   molecular weight of dry air, of water



| | |
|---|---|
| $N, N'$ | number concentration of cloud droplets, of ambient aerosol particles |
| $p$ | pressure |
| $r, r_c$ | radius of cloud droplet, of dry aerosol particle |
| $R$ | universal gas constant |
| $R_B, R_J$ | radius of air bubble, of convective jet |
| $S$ | supersaturation |
| $S_{eq}$ | droplet equilibrium supersaturation |
| $T (T')$ | temperature of air parcel (ambient air) |
| $V$ | parcel updraft velocity |
| $v, v'$ | droplet volumes |
| $w_L$ | mixing ratio of liquid water in parcel |
| $w_v (w_v')$ | mixing ratio of water vapor in parcel (in environment) |
| $\kappa$ | hygroscopicity parameter |
| $\mu$ | entrainment rate |
| $\rho_a, \rho_w$ | density of dry air, of water |
| $\sigma_w$ | droplet surface tension |

*Additional Formulae*

$$G = \left[ \frac{\rho_w RT}{e_s D_v' M_w} + \frac{L \rho_w}{k_a' T} \left( \frac{L M_w}{RT} - 1 \right) \right]^{-1} \tag{A1}$$

where the modified diffusivity ($D_v'$) and thermal conductivity ($k_a'$) of water vapor in air account for non-continuum effects (Seinfeld and Pandis, 2006) and are described as follows

$$D_v' = \frac{D_v}{1 + \frac{D_v}{a_c r} \sqrt{\frac{2 \pi M_w}{RT}}} \tag{A2}$$

$$k_a' = \frac{k_a}{1 + \frac{k_a}{a_T r \rho_a c_p} \sqrt{\frac{2 \pi M_a}{RT}}} \tag{A3}$$

5 where the thermal accommodation coefficient ($a_T$) is taken as 0.96 (Nenes et al., 2001). Additional sensitivity tests of CDNC to $a_T$, ranging from 0.1 to 1 (Shaw and Lamb, 1999), were conducted and the resulting droplet concentrations indicate little sensitivity to this input parameter (not shown here).

The hygroscopicity parameter ($\kappa$) is adopted to characterize aerosol chemical composition on CCN activity according to $\kappa$-Köhler theory (Petters and Kreidenweis, 2007). $S_{eq}$ for droplets in the $i^{th}$ bin ($i = 1, 2,..., nbin$) can be written as

$$S_{eq} = \frac{r_i^3 - r_{c,i}^3}{r_i^3 - r_{c,i}^3 (1 - \kappa_i)} exp \left( \frac{2 M_w \sigma_w}{RT \rho_w r_i} \right) - 1 \tag{A4}$$





where $r_{c,i}$ and $r_i$ are the radius of the dry aerosol particle and the corresponding growing droplet, respectively. Droplet surface tension $(\sigma_w)$ is a function of the parcel temperature (Pruppacher and Klett, 1997).

$$\alpha = \frac{gM_w L}{c_p R T^2} - \frac{gM_a}{RT} \qquad (A5)$$

$$\gamma = \frac{pM_a}{e_s M_w} + \frac{M_w L^2}{c_p R T^2} \qquad (A6)$$

Liquid water content (g m$^{-3}$):

$$LWC = \frac{4\pi}{3}\rho_w \sum_{i=1}^{bins} N_i r_i^3 \qquad (A7)$$

## Appendix B

### 1. Sensitivity to environmental conditions

To account for the uncertainties associated with the environmental condition from WRF and examine its impact on cloud formation, two additional simulations were conducted with modified profiles of temperature and humidity at the lowest 2 km above CBH (1,270 m), as displayed in Fig. B1. In the first simulation, we adjusted the original lapse rate (-4.1 °C km$^{-1}$ from the WRF sounding, Fig. 10b) to -7 °C km$^{-1}$ ($\Gamma_1$) for 1,270–2,200 m. In the second one, specific humidity (q) in the environment was increased by 5% for 1,270–3,200 m together with the adjusted temperature profile in the first run. In both simulations, the lapse rate for 2,200–3,200 m was changed to -4 °C km$^{-1}$ to keep the ambient temperature below CBH and above 3,200 m unchanged. As expected, deeper clouds are formed in modified environments representing conditionally unstable atmosphere. A slight increase in specific humidity has little influence on the maximum supersaturation formed near cloud base. Consequently, its effect on droplet concentrations is also negligible due to the slightly increased maximum supersaturation (not shown here). It is expected that LWC is significantly enhanced and droplet growth is faster under the environmental condition of fast cooling and moist air.

### 2. Sensitivity to hygroscopicity

Another key element in the condensation process is the hygroscopic property that governs the influence of aerosol chemical composition on CCN activity. To account for its temporal variability observed during IPHEx, a $\kappa$ value varying from 0.1–0.4 (within the typical range measured at the surface site, see Figs. 4a and S8c) is applied uniformly for all particle sizes. As noted from Fig. B2, simulated profiles of supersaturation and total CDNC exhibit a weak dependence on the hygroscopicity that a slightly decrease in maximum supersaturation and a slightly increase in total CDNC are associated with more hygroscopic aerosols. Predicted droplet spectra at three altitudinal levels (1,500 m: solid line, 1,600 m: dotted line, and 1,700 m: dashed line) also show little sensitivity to the variations in $\kappa$. As discussed in Sect. 3.1, hygroscopic properties of aerosols have been found to vary with particle sizes. Potential uncertainties might remain by assuming a constant $\kappa$, but its





variation with droplet sizes is not addressed in the current study. We should also note the hygroscopicity derived from surface measurements may not be representative for aerosols beneath the cloud (Pringle et al., 2010). However, the vertical variability of aerosol hygroscopicity is not taken into account in this study.

**3. Sensitivity to initial updraft velocity**

Cloud dynamics also play a crucial role in the microphysical evolution of cumulus clouds. One major parameter in the cloud dynamical field is the updraft velocity. In accordance with the observed vertical velocities from the aircraft and the W-band radar (see Fig. 6b), a reasonable variability in the initial updraft velocity at cloud base is introduced to assess its effects on the parcel supersaturation and cloud droplet concentrations, as shown in Fig. B3. By varying the initial updraft in a

range of 0.1–1.5 m s$^{-1}$, simulated results display similar vertical velocities at the observation levels, which are still higher than the measured range (not shown here). As expected, slight increases in maximum supersaturation are resulted from larger initial updraft velocities, thus leading to slight enhancement of total droplet numbers. The simulated spectra show a slightly shift towards larger drop sizes due to weaker updrafts, which allow more time for cloud droplets to grow in a rising parcel.

**Data availability:** The IPHEx data are accessible at Global Hydrology Resource Center (GHRC) Distributed Active Archive Center (https://ghrc.nsstc.nasa.gov/home/field-campaigns/iphex).

**Competing interests:** The authors declare that they have no conflict of interest.

**Acknowledgements**: This work was supported in part by NASA grant NNX16AL16G with Barros and NSF Rapid Response Research (RAPID) Collaborative IPHEx grant with Barros (1442039) and Petters (1442056). Yajuan Duan developed the Duke Cloud Parcel Model (DCPM) and conducted the modelling study under the guidance of Ana Barros. Markus Petters was the lead researcher operating the SMPS/CCN system during IPHEx, and provided level-2 datasets of SMPS/CCN

measurements (Sect. 3.1). Barros and Duan wrote the manuscript, and Petters provided comments. The authors thank the UND Citation flight scientists, in particular Michael Poellot, Andrew Heymsfield, and David J. Delene for the flight data and advice with airborne data analysis, Si-Chee Tsay and Adrian Loftus for the deployment and operation of the ACHIEVE instruments and W-band radar calibrated data, Anna M. Wilson for the deployment and maintenance of Duke's H2F (Haze to Fog) mobility facility (including the PCASP and raingauges) and corresponding data collection and analysis, and Andrew

Grieshop for loaning the X-Ray neutralizer for the duration of the study. We also thank Kyle Dawson and John Hader for operating the SMPS/CCN system in the field and Kyle Dawson for help with the processing of SMPS/CCN data set (Sect. 3.1). We also knowledge computing resources from Yellowstone (*ark:/85065/d7wd3xhc*) at NCAR (allocated to the first author) used for the WRF simulations. The authors are especially grateful to Neil Carpenter from the Maggie Valley Sanitary District for his support of IPHEx activities.




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



**Table 1.** Cloud parcel models with detailed microphysics from the literature and in this study (Duke CPM). NA denotes information is not described in the reference paper.

| Parcel model | Binning | Condensation | Coalescence | Entrainment | Numerics |
|---|---|---|---|---|---|
| Abdul-Razzak et al. (1998) | Discrete | Leaitch et al. (1986) | Not included | No included | LSODE solver (Hindmarsh, 1983) |
| Cooper et al. (1997) | Moving discrete | Fukuta and Walter (1970) | Modified Kovetz and Olund (1969) | Not included | Fifth-order Runge-Kutta (adaptive-size) |
| Flossmann et al. (1985) | Discrete | Pruppacher and Klett (1978) | Berry and Reinhardt (1974) | Lateral homogeneous bubble model | NA |
| Jacobson and Turco (1995) | Hybrid discrete | Jacobson and Turco (1995) | Jacobson et al. (1994) | Not included | SMVGEAR (Jacobson and Turco, 1994) |
| Kerkweg et al., (2003) | Discrete | Pruppacher and Klett (1997) | Bott (2000) | Lateral homogeneous bubble model | NA |
| Nenes et al. (2001; 2002) | Moving discrete | Pruppacher and Klett (1997); Seinfeld and Pandis (1998) | Not included | Not included | LSODE solver (Hindmarsh, 1983) |
| Pinsky and Khain (2002) | Moving discrete | Pruppacher and Klett (1997) | Bott (1998); turbulent effect on drop collision | Not included | NA |
| Snider et al. (2003) | Discrete | Zou and Fukuta (1999) | Not included | Not included | NA |
| *Duke CPM* | Moving discrete | Pruppacher and Klett (1997); Seinfeld and Pandis (2006) | Bott (1998); turbulent effect on drop collision | Lateral homogeneous bubble/jet model | Fifth-order Runge-Kutta (adaptive-size) |





**Table 2. Lognormal fit parameters characterizing the aerosol number distribution of four modes. Note N = total number of aerosol particles per cm³; Dg = geometric mean diameter (µm); $\sigma_g$ = geometric standard deviation for each mode. $N_{surf}$ and $N_{CBH}$ represent total aerosol number concentrations at the surface and cloud base height (CBH: 1,270 m), respectively.**

| Mode # | $N_{surf}$ (cm⁻³) | $N_{CBH}$ (cm⁻³) | $D_g$ (µm) | $\sigma_g$ |
|--------|-------------------|------------------|------------|------------|
| 1 | 1401.9 | 393.7 | 0.076 | 1.63 |
| 2 | 415.7 | 116.8 | 0.195 | 1.35 |
| 3 | 0.3 | 0.084 | 0.75 | 1.3 |
| 4 | 0.3 | 0.084 | 2.2 | 1.4 |





**Table 3. Evaluation of the predicted CDNC from simulations using various condensation coefficients against the averaged observation from the CDP.**

| Condensation coefficient | Prediction[a] (cm$^{-3}$) | Difference[b] (%) |
|:---:|:---:|:---:|
| 0.002 | 402.7 | 15.3 |
| 0.005 | 385.8 | 10.4 |
| 0.01 | 354.0 | 1.3 |
| 0.015 | 328.5 | -6 |
| 0.03 | 281.0 | -19.6 |
| 0.06 | 242.1 | -30.7 |

[a]The averaged CDNC in the predictions between 1,500 m and 1,600 m AGL.

[b]Difference (%) = 100×(Prediction - Observation)/Observation. Note observation = 349.4 cm$^{-3}$, the average of five CDNC measurements
5   between 1,500 m and 1,600 m AGL.



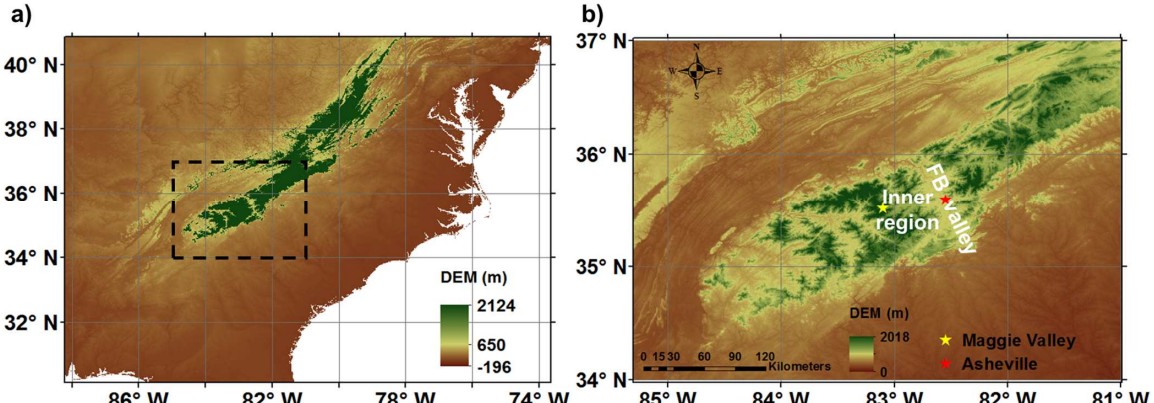

**Figure 1: a) Study region of the IPHEx campaign in the SAM (highlighted in the black box), as shown in context of a large scale map of the southeastern United States. (b) Topographic map of the SAM including the two ground-based IPHEx observation sites referred to in this study. FB valley denotes French Broad valley.**




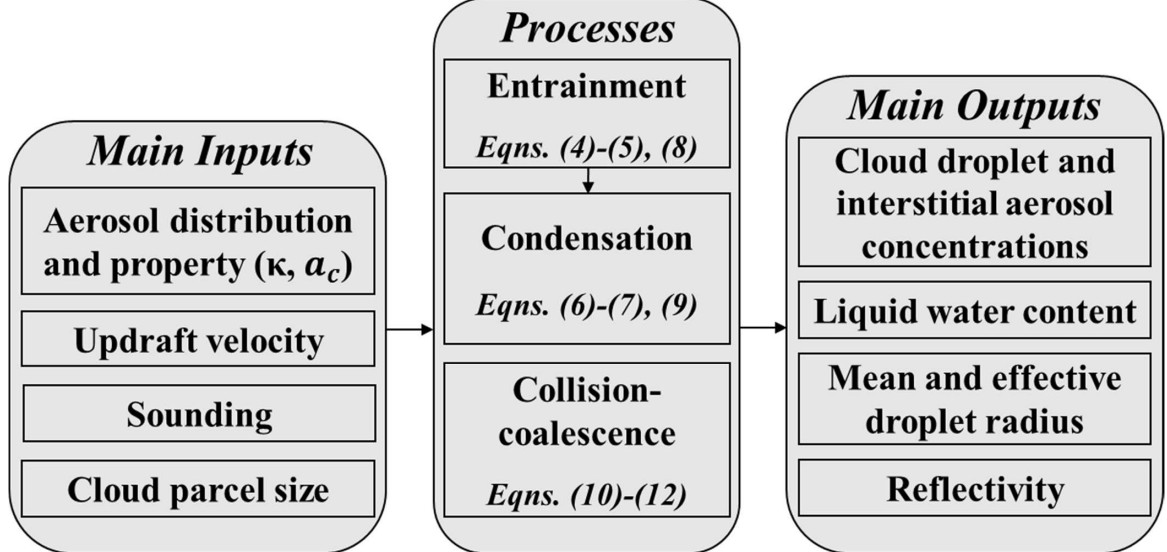

**Figure 2: Flowchart of the main inputs, microphysical processes, and main outputs of the DCPM. Equation numbers refer to formulae in Sect. 2.**



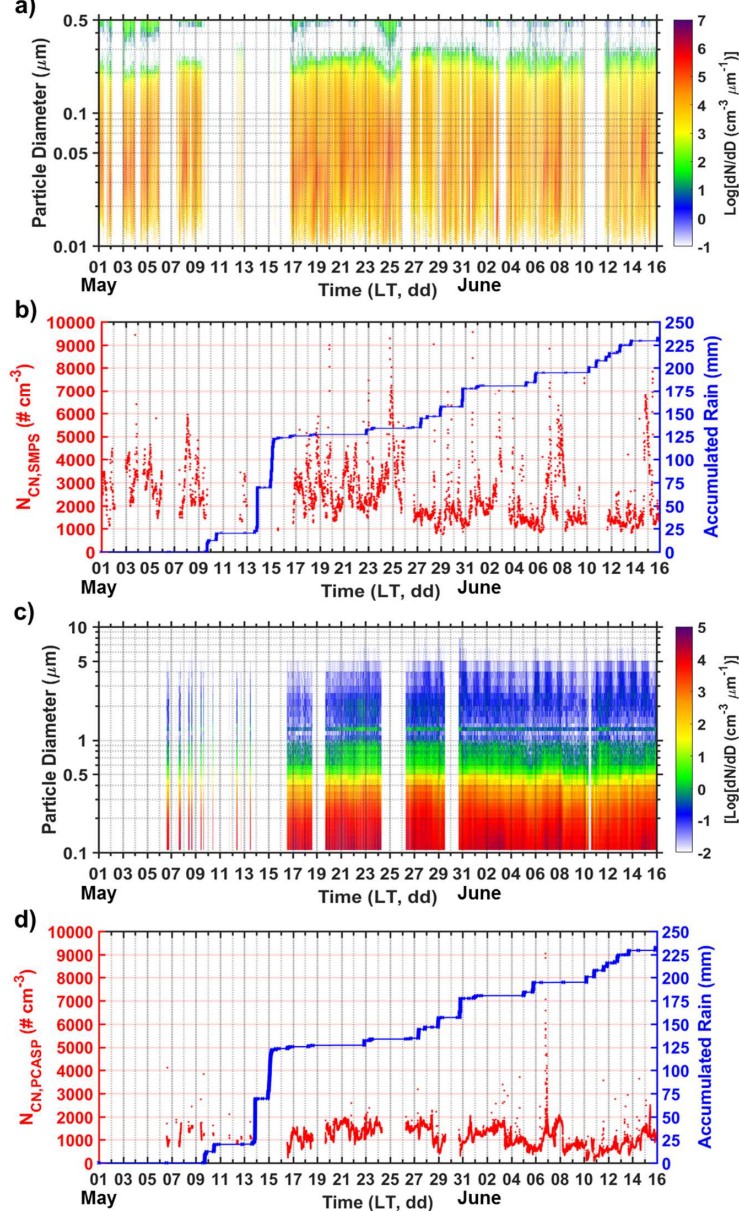

**Figure 3: Time series of dry aerosol size distribution and total number concentration ($N_{CN}$) from the SMPS (a, b) and PCASP (c, d), respectively, measured at MV during the IPHEx IOP. Discontinuities in the data are associated with delayed installation (PCASP), rainfall occurrences, and occasional instrument malfunction.**


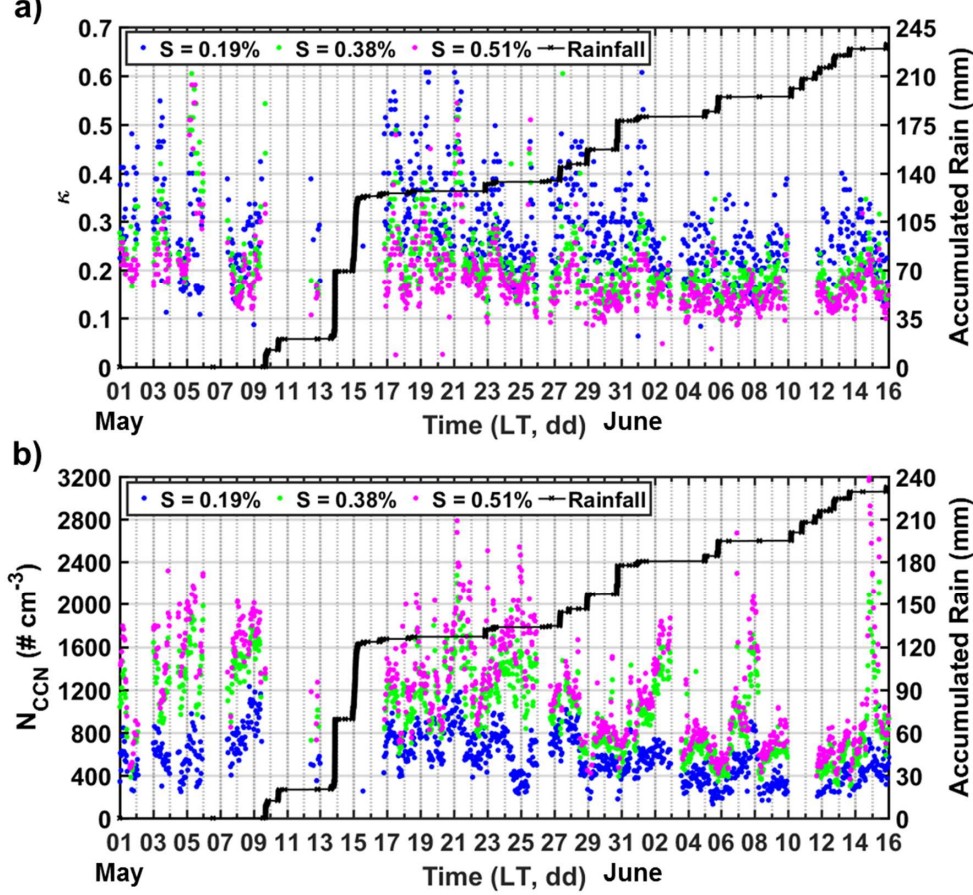

**Figure 4:** Time series of hygroscopicity parameter ($\kappa$, a) and CCN concentration ($N_{CCN}$, b) at three supersaturation levels, measured at MV during the IPHEx IOP. Discontinuities in the data are associated with rainfall occurrences and occasional instrument malfunction.





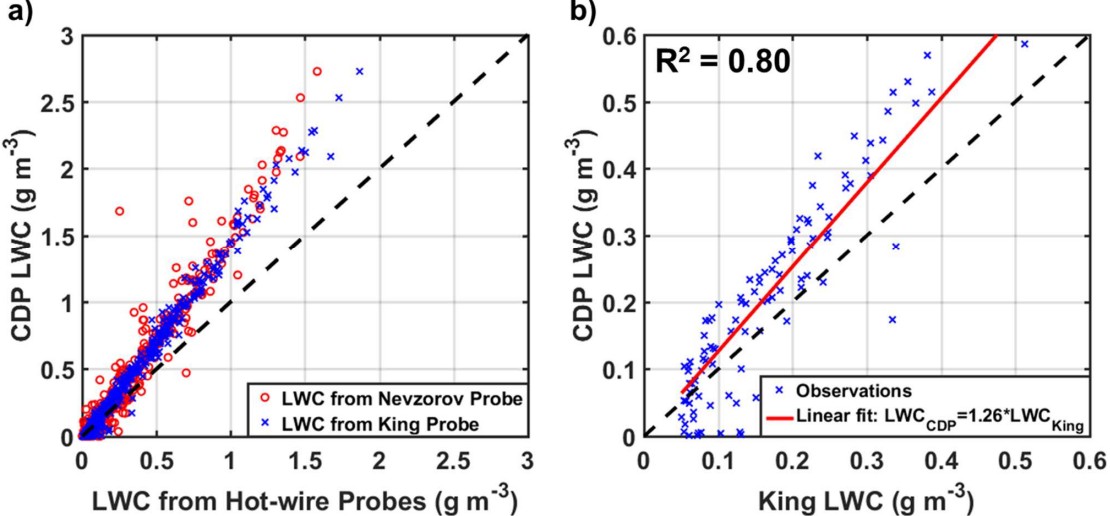

**Figure 5: a) Scatterplot for LWC from the CDP and two hot-wire probes (the King and Nevzorov probes), sampled during the 12 June flight. b) LWC observations from the CDP and the King probe during the first horizontal cloud transect on the same day are fitted by a linear regression (represented by the red line) with coefficient of determination $R^2 = 0.80$.**





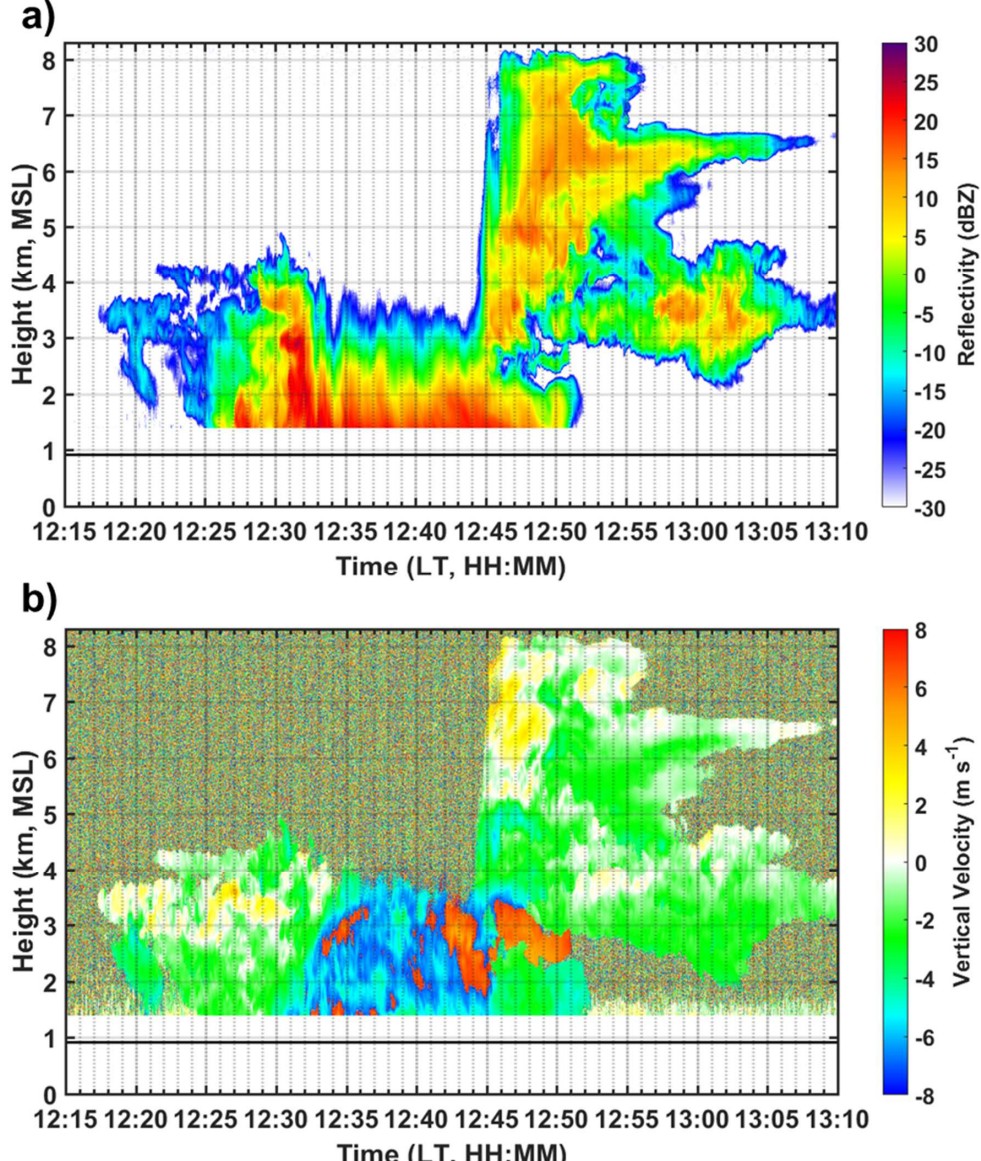

**Figure 6: Vertical profiles of W-band reflectivity (a) and vertical velocity (b) of cumulus congestus clouds observed at MV on 12 June 2014. The horizontal line depicts the elevation level of the MV supersite (~ 925 m MSL).**



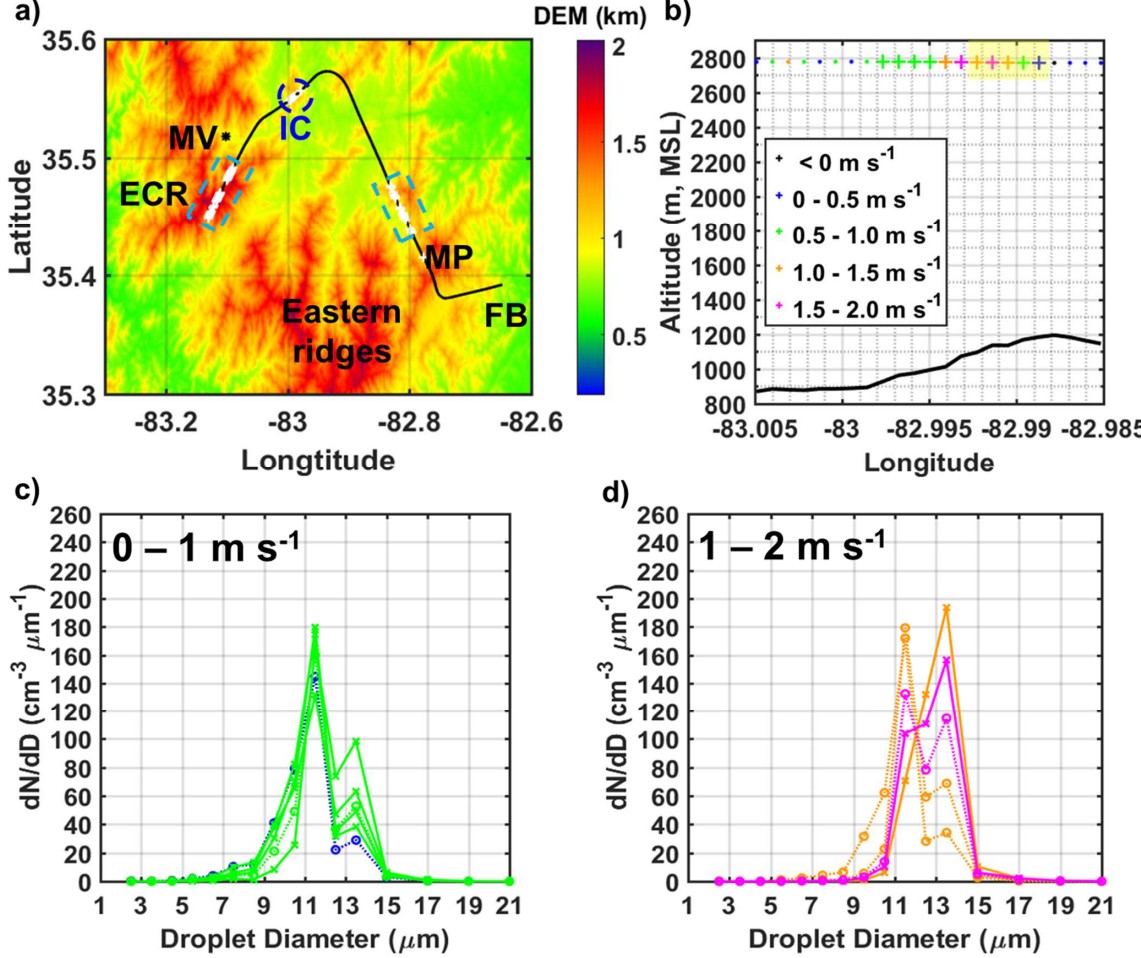

Figure 7: a) Lowest cloud transect of the UND Citation flight track on 12 June 2014. The in-cloud observations are identified as white plus signs, and MV is marked by the black asterisk. For left to right in the map, ECR denotes Eastern Cherokee reservation, MP denotes Mount Pisgah, and FB denotes French Board valley. b) Velocity variations of the targeted in-cloud region, denoted by IC in (a). Coloured plus signs indicate updraft velocities of the in-cloud samples, collected at 1-Hz (~ 90 m in flight distance) resolution. Cloud droplet concentrations of the in-cloud samples in IC (b) with low (0–1 m s⁻¹) and high (1–2 m s⁻¹) updrafts are shown in (c) and (d), respectively. The updraft velocity of each sample is indicated by its colour, referring to the range in the legend of (b). Dotted lines with circle markers represent the droplet spectra in the reference sub-region within IC, as shaded in (b).



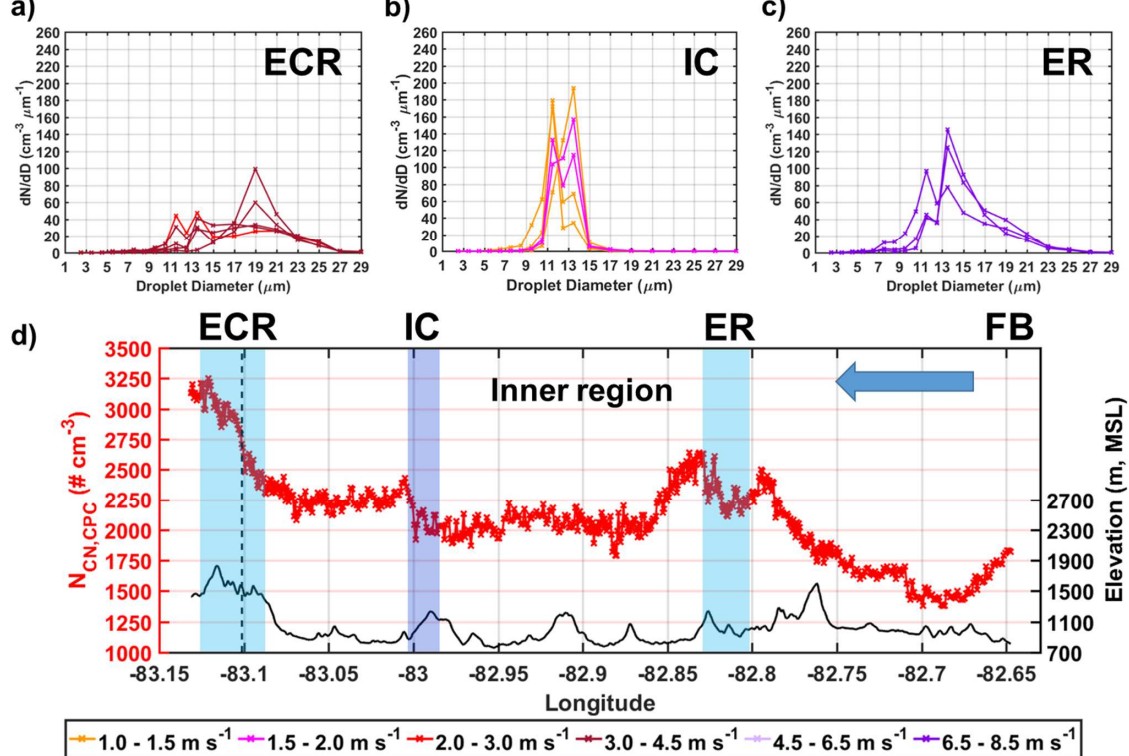

Figure 8: Cloud droplet concentrations at the updraft core of clouds near the Eastern Cherokee reservation (ECR, a), within the targeted in-cloud region (IC, b), and near the foothills of the eastern ridges (ER) over the inner region (c), respectively. Their locations can be referred to Fig. 7a. The updraft velocity of each sample is denoted by its colour. d) Background aerosol concentrations from the CPC abroad the UND Citation during the first horizontal leg (see flight track in Fig. 7a, and the flight direction is indicated by the blue arrow here). The blue shaded areas correspond to cloudy regions in (a)–(c), also as highlighted in the dark blue circle and light blue boxes in Fig. 7a. The terrain elevation is represented by the black line and FB denotes French Board valley.




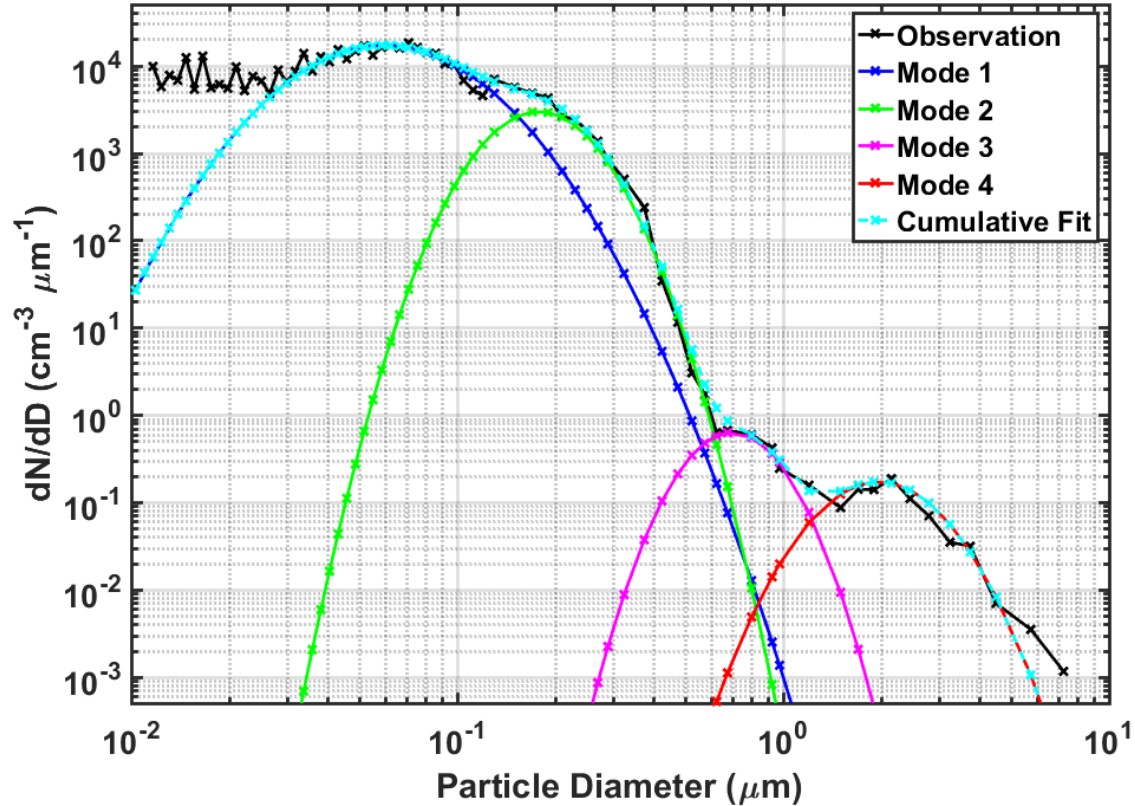

**Figure 9: Mean surface aerosol size distribution fitted by four lognormal functions. Observations are merged from the SMPS and PCASP, and are averaged during the first 10 mins (12:14 LT – 12:24 LT) of the 12 June flight. Fitted parameters (total number concentration, geometric mean diameter, and geometric standard deviation) for each mode are summarized in Table 2.**



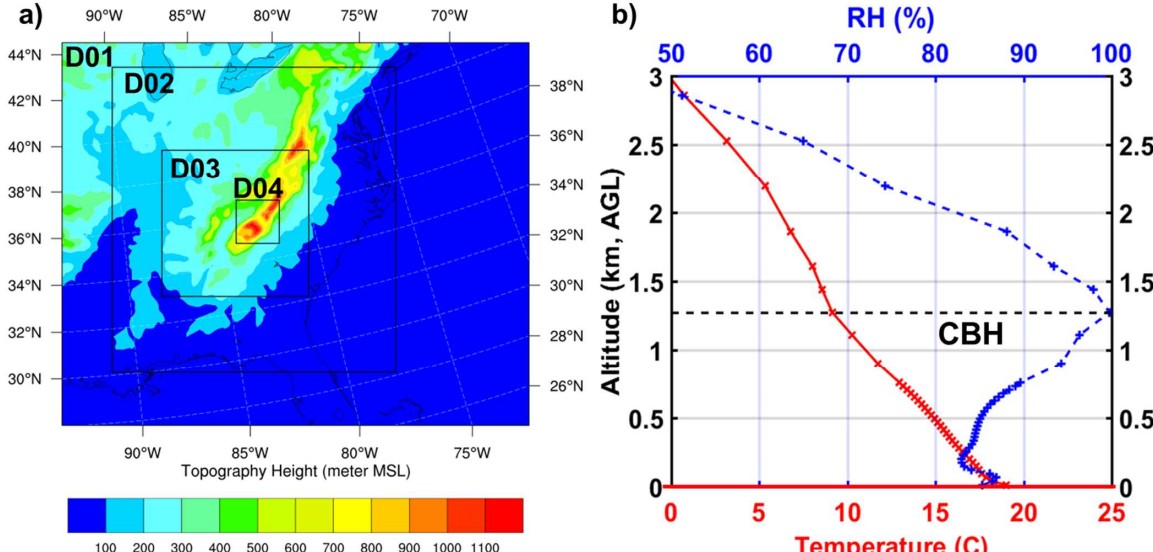

**Figure 10: a)** WRF model configuration of four one-way nested domains at 15-, 5-, 1.25-, 0.25-km grid resolution, respectively. **b)** Vertical profile of temperature (red solid line) and relative humidity (dashed blue line) from the spatially-averaged WRF sounding columns at IC (see its location in Fig. 7a). The horizontal dashed line depicts CBH = 1,270 m AGL.







**Figure 11: Sensitivity of the updraft velocity (a), supersaturation (b), total drop concentration (c), and LWC (d) to the variations in the condensation coefficient ($a_c$) as compared to the airborne observations, marked by the different black symbols denoting the ranges of their updraft velocities (triangles: 0–0.5 m s$^{-1}$, squares: 0.5–1.0 m s$^{-1}$, pentagrams: 1–1.5 m s$^{-1}$, hexagrams: 1.5–2.0 m s$^{-1}$). The horizontal dashed line depicts CBH.**





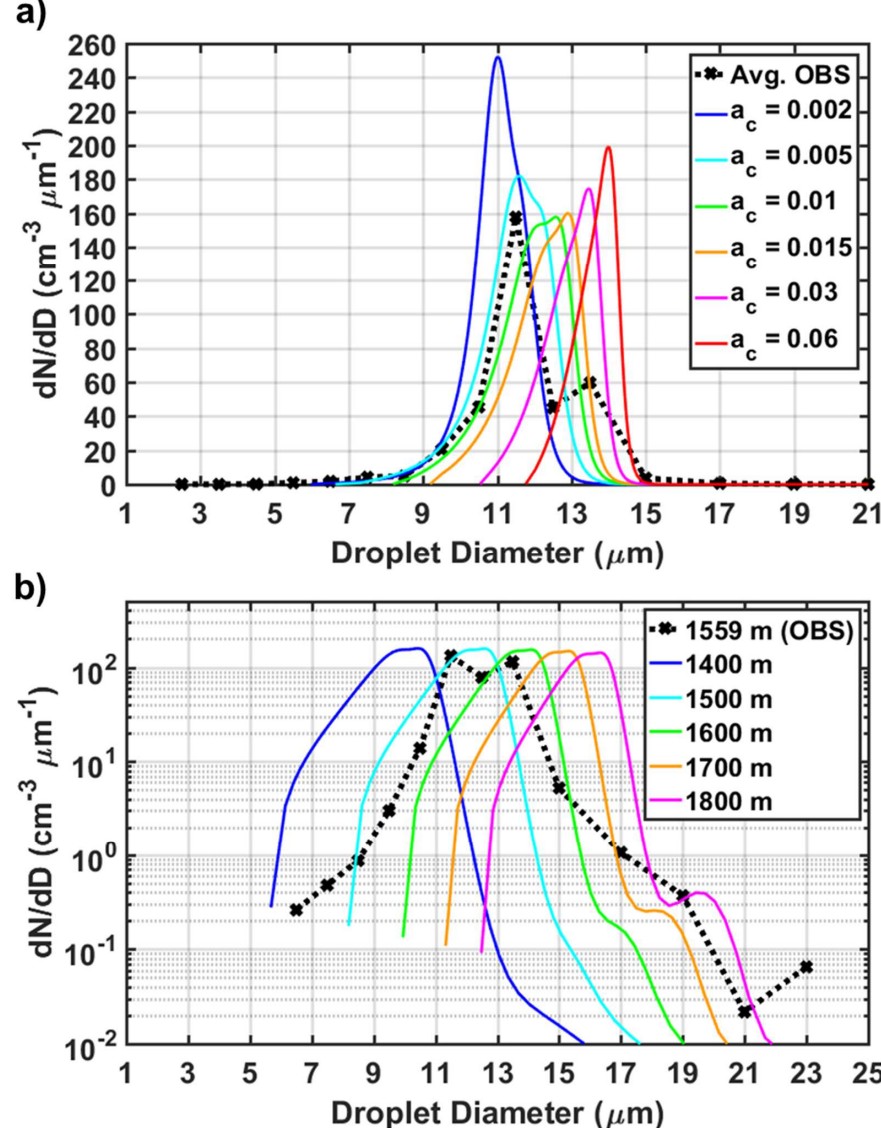

**Figure 12: a) Sensitivity of simulated droplet spectra at 1,500 m (solid lines) to the variations in $a_c$. The black dotted line reflects the average of five droplet spectra observed by the CDP (dotted lines with circle markers in Figs. 7c and d) between 1,500 m and 1,600 m AGL. b) Simulated evolution of cloud droplet spectra at 1,400 m, 1,500 m, 1,600 m, 1,700 m, and 1,800 m altitude assuming $a_c$ = 0.01. The black dotted line denotes the observed droplet spectrum at 1,559 m that has similar total CDNC and LWC as the simulation with $a_c$ = 0.01 at the same altitude.**





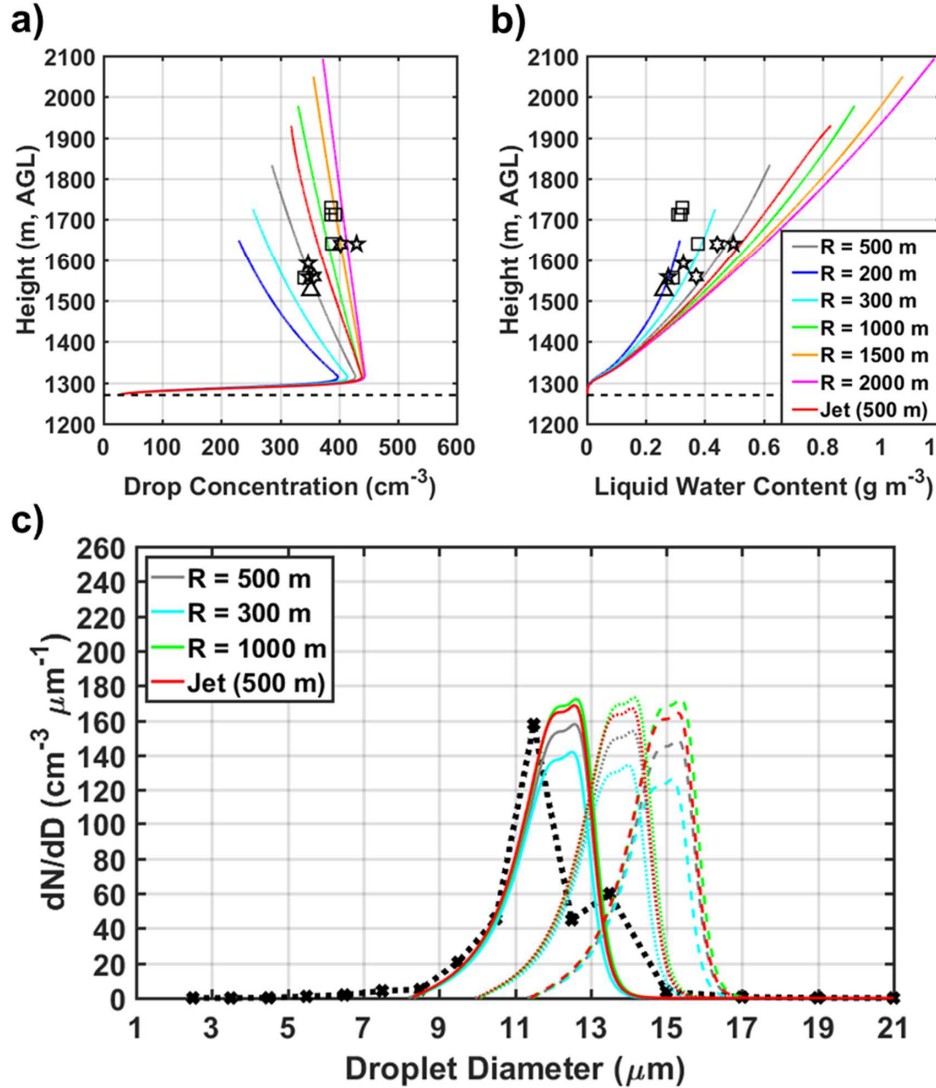

**Figure 13: Sensitivity of the total drop concentration (a) and LWC (b) to the variations in the initial parcel radius (R) considering lateral entrainment as a bubble model and a jet model. In (a) and (b), the airborne observations are marked by different black symbols, denoting the ranges of their updraft velocities (triangles: 0–0.5 m s[-1], squares: 0.5–1.0 m s[-1], pentagrams: 1–1.5 m s[-1], hexagrams: 1.5–2.0 m s[-1]), and the horizontal dashed line depicts CBH. c) Predicted droplet spectra at three altitudinal levels (1,500 m: solid line, 1,600 m: dotted line, and 1,700 m: dashed line) using two parameterization schemes for lateral entrainment: the bubble model with R = 500 m (base case, grey lines), R = 300 m (cyan lines), and R = 1,000 m (green lines); the jet model with R = 500 m (red lines). The black dotted line reflects the average of five droplet spectra observed by the CDP (dotted lines with circle markers in Figs. 7c and d) between 1,500 m and 1,600 m AGL.**





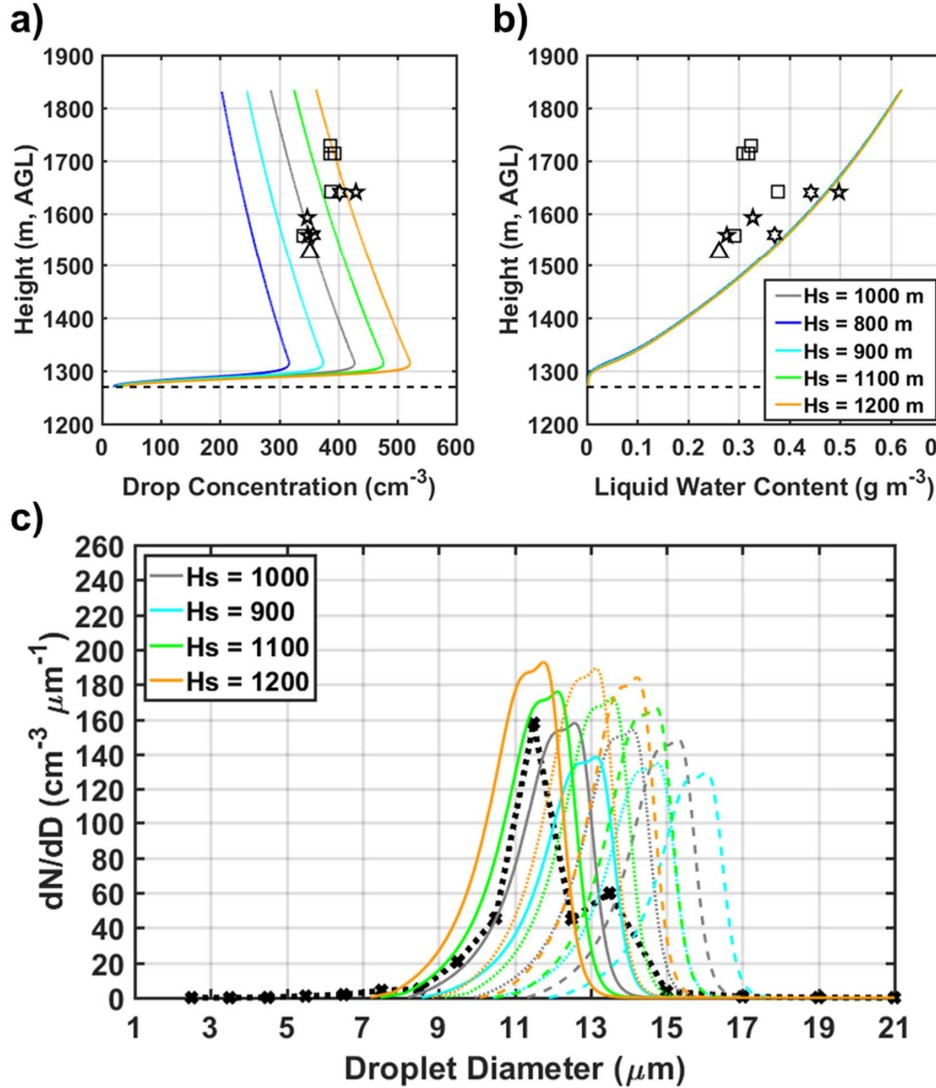

**Figure 14: Sensitivity of the total drop concentration (a), LWC (b), and droplet spectra (c) at three altitudinal levels (1,500 m: solid line, 1,600 m: dotted line, and 1,700 m:dashed line) to the variations in initial aerosol concentrations at cloud base, as represented by different values of the scale height (H$_S$). In (a) and (b), the airborne observations are marked by different black symbols, denoting the ranges of their updraft velocities (triangles: 0–0.5 m s$^{-1}$, squares: 0.5–1.0 m s$^{-1}$, pentagrams: 1–1.5 m s$^{-1}$, hexagrams: 1.5–2.0 m s$^{-1}$), and the horizontal dashed line depicts CBH. The black dotted line reflects the average of five droplet spectra observed by the CDP (dotted lines with circle markers in Figs. 7c and d) between 1,500 m and 1,600 m AGL.**



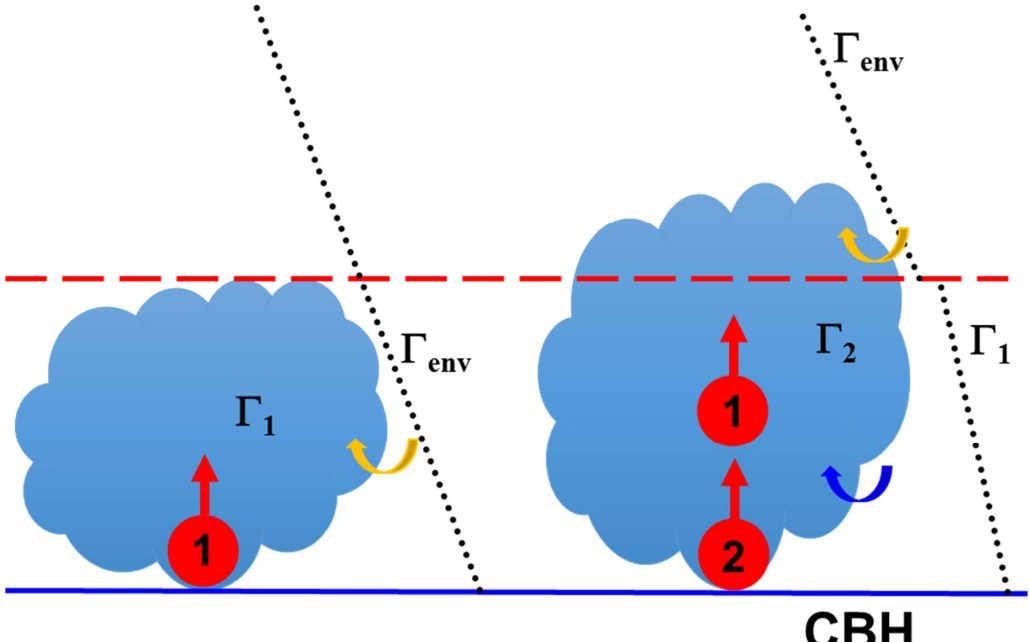

**Figure 15: Conceptual representation of the double-parcel simulations. The behaviour of the second parcel is determined by the new environment ($\Gamma_1$) established by the first parcel. During the ascent of the second parcel, interstitial aerosols left behind by the first one are entrained (indicated by the blue curved arrows). When the second parcel rises above the maximum height that the first parcel has reached (indicated by the red dashed line), its behaviour is determined by the initial environment ($\Gamma_{env}$) and aerosol particles from the environment are entrained (indicated by the yellow curved arrows). The order of each parcel is denoted inside the red circle and the blue solid line marks CBH.**



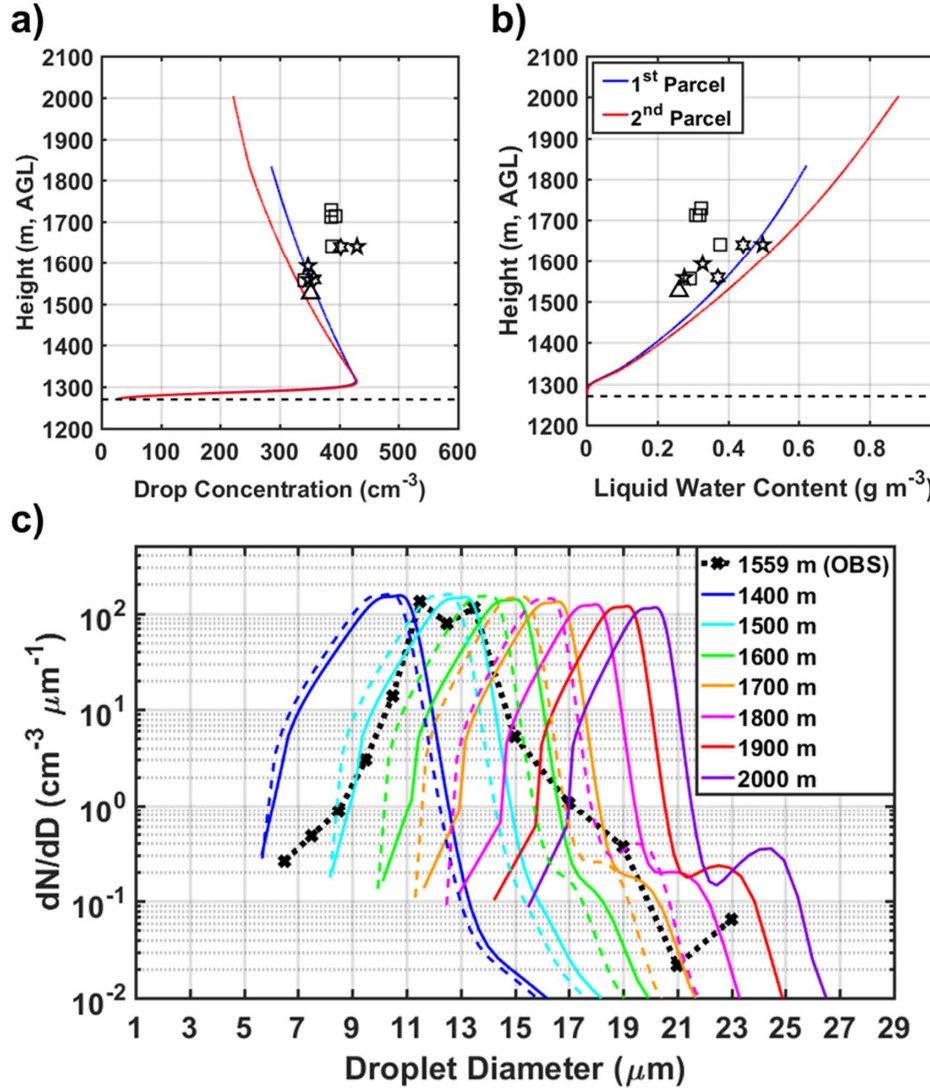

**Figure 16: Vertical profiles of the simulated total drop concentration (a) and LWC (b) for the first and second parcels. In (a) and (b), the airborne observations are marked by different black symbols, denoting the ranges of their updraft velocities (triangles: 0–0.5 m s⁻¹, squares: 0.5–1.0 m s⁻¹, pentagrams: 1–1.5 m s⁻¹, hexagrams: 1.5–2.0 m s⁻¹), and the horizontal dashed line depicts CBH. c) Simulated evolution of cloud droplet spectra for the first (represented by the dashed lines) and second (represented by the solid lines) parcels at different altitudinal levels. The black dotted line denotes the observed droplet spectrum at 1,559 m that has similar total CDNC and LWC as the simulated spectrum in the first parcel at the same altitude.**



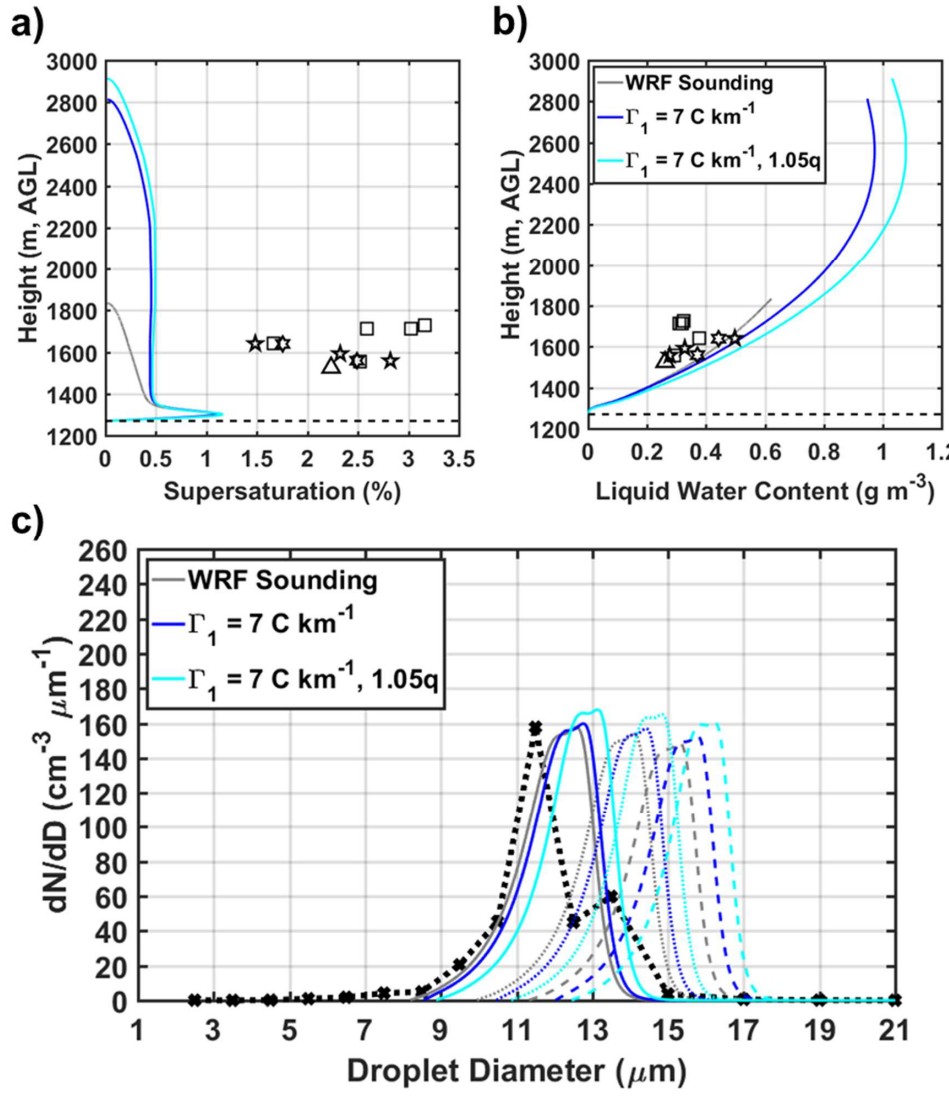

**Figure B1: Vertical profiles of the supersaturation (a) and LWC (b) for simulations with the original WRF sounding (grey lines), modified ambient temperature (blue lines), and modified ambient temperature and humidity (cyan lines). In (a) and (b), the airborne observations are marked by different black symbols, denoting the ranges of their updraft velocities (triangles: 0–0.5 m s⁻¹, squares: 0.5–1.0 m s⁻¹, pentagrams: 1–1.5 m s⁻¹, hexagrams: 1.5–2.0 m s⁻¹), and the horizontal dashed line depicts CBH. c) Predicted droplet spectra at three altitudinal levels (1,500 m: solid line, 1,600 m: dotted line, and 1,700 m: dashed line) to the variations in the environmental conditions modified from the WRF sounding. The black dotted line reflects the average of five droplet spectra observed by the CDP (dotted lines with circle markers in Figs. 7c and d) between 1,500 m and 1,600 m AGL.**

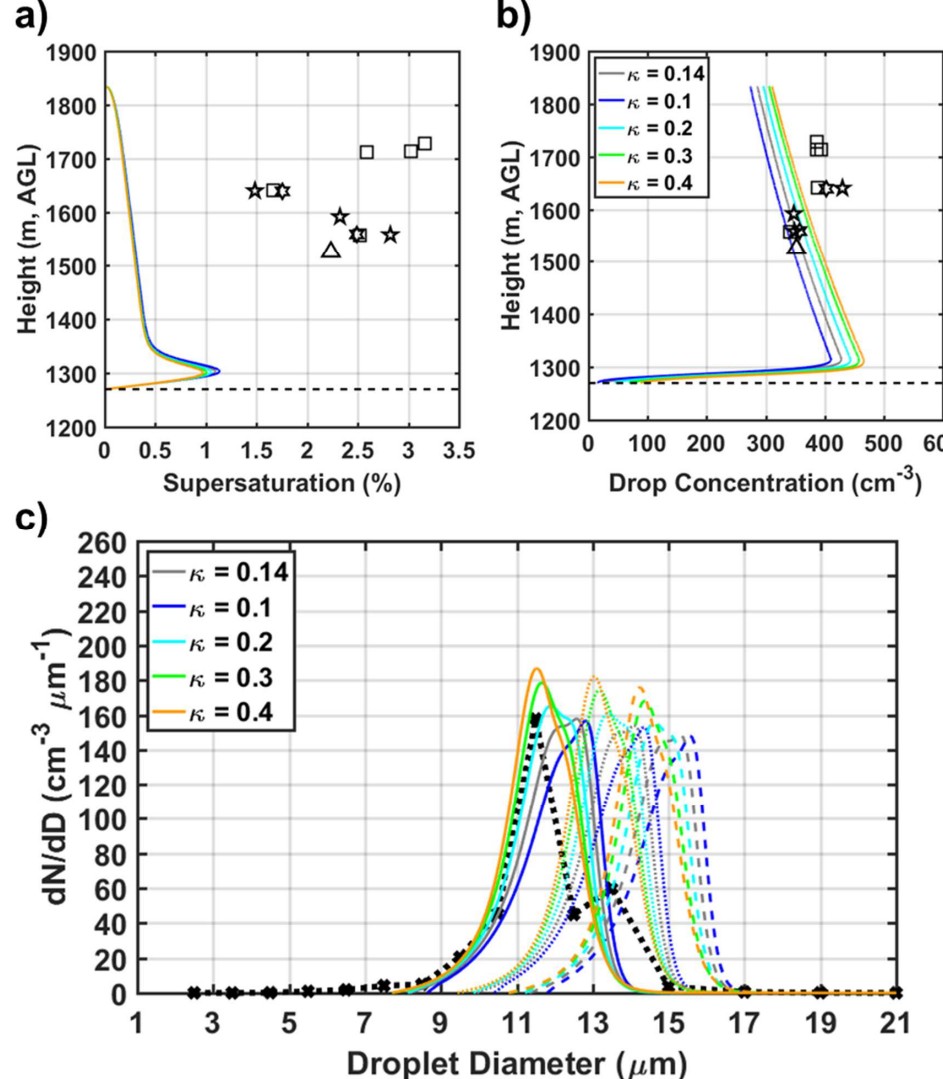

**Figure B2: Sensitivity of the supersaturation (a) and total drop concentration (b) to variations in hygroscopicity parameter (κ). In (a) and (b), the airborne observations are marked by different black symbols, denoting the ranges of their updraft velocities (triangles: 0–0.5 m s$^{-1}$, squares: 0.5–1.0 m s$^{-1}$, pentagrams: 1–1.5 m s$^{-1}$, hexagrams: 1.5–2.0 m s$^{-1}$), and the horizontal dashed line**
5 **depicts CBH. c) Predicted droplet spectra at three altitudinal levels (1,500 m: solid line, 1,600 m: dotted line, and 1,700 m: dashed line) to the variations in κ. The black dotted line reflects the average of five droplet spectra observed by the CDP (dotted lines with circle markers in Figs. 7c and d) between 1,500 m and 1,600 m AGL.**



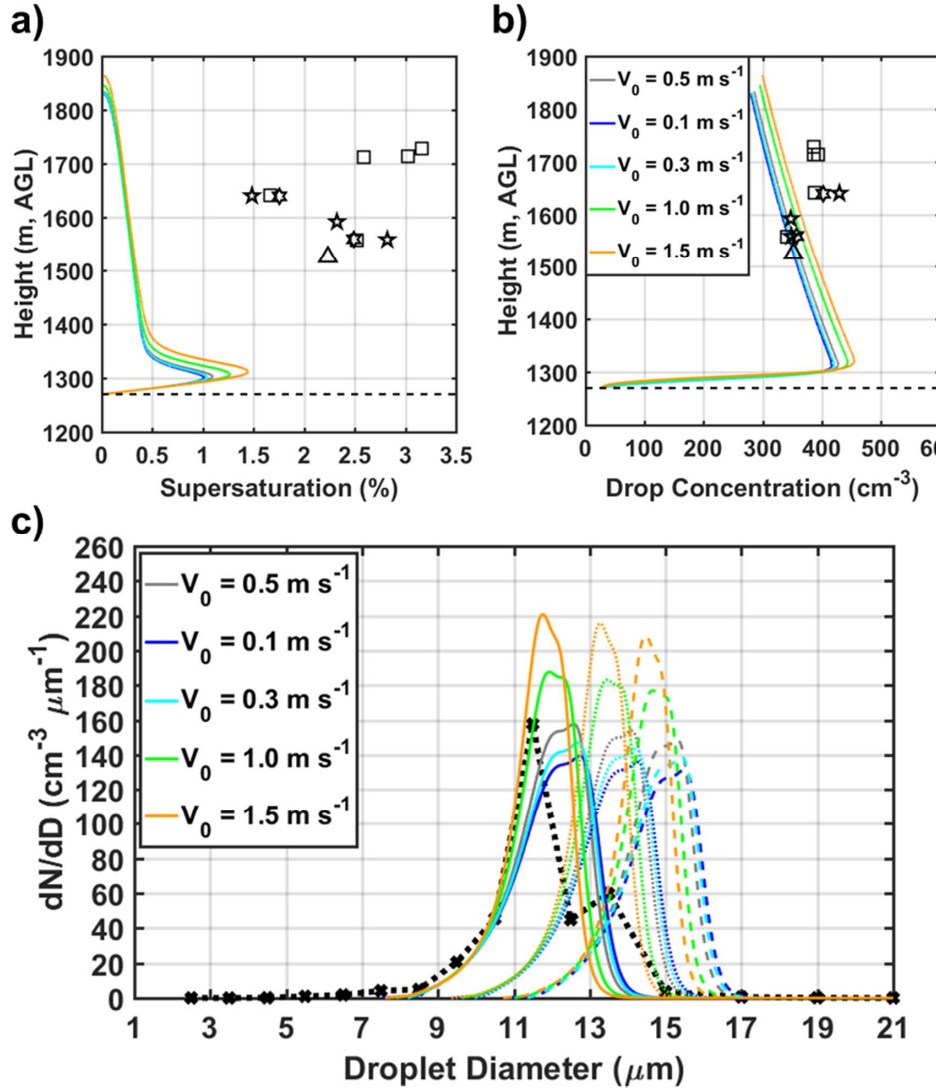

**Figure B3: Sensitivity of the supersaturation (a), total drop concentration (b), and droplet spectra (c) at three altitudinal levels (1,500 m: solid line, 1,600 m: dotted line, and 1,700 m:dashed line) to the variations in the initial updraft velocity ($V_0$) at cloud base. In (a) and (b), the airborne observations are marked by different black symbols, denoting the ranges of their updraft velocities (triangles: 0–0.5 m s$^{-1}$, squares: 0.5–1.0 m s$^{-1}$, pentagrams: 1–1.5 m s$^{-1}$, hexagrams: 1.5–2.0 m s$^{-1}$), and the horizontal dashed line depicts CBH. The black dotted line in (c) reflects the average of five droplet spectra observed by the CDP (dotted lines with circle markers in Figs. 7c and d) between 1,500 m and 1,600 m AGL.**





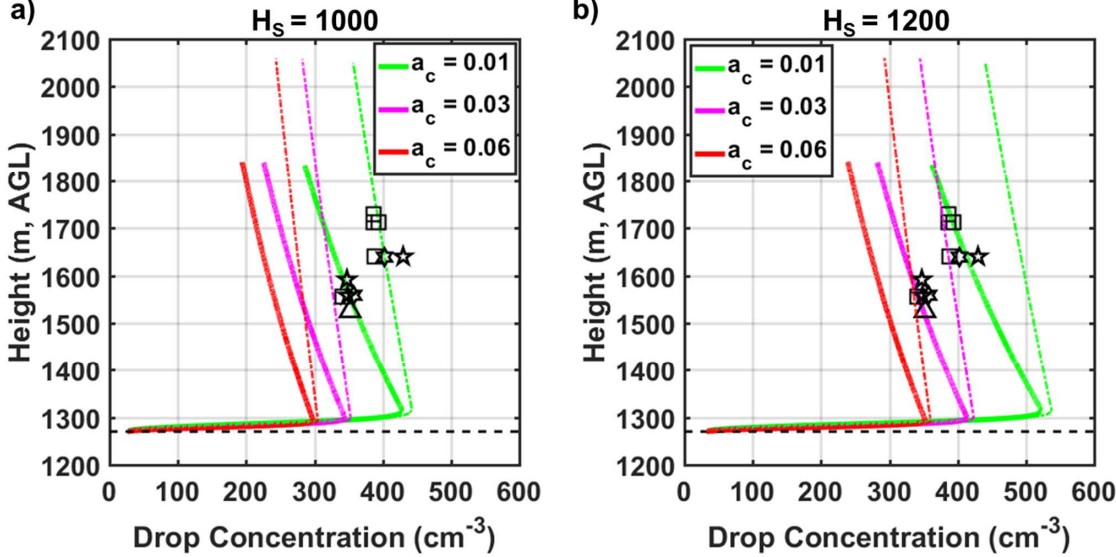

**Figure B4: Sensitivity of the total cloud drop concentration to the variations in condensation coefficient and entrainment strength (strong: R = 500 m, solid thick lines; weak: R = 1,500 m, dash-dot thin lines) assuming different initial aerosol concentrations at cloud base (a: $H_S$ = 1,000 m; b: $H_S$ = 1,200 m). The airborne observations are marked by different black symbols, denoting the ranges of their updraft velocities (triangles: 0–0.5 m s$^{-1}$, squares: 0.5–1.0 m s$^{-1}$, pentagrams: 1–1.5 m s$^{-1}$, hexagrams: 1.5–2.0 m s$^{-1}$). The horizontal dashed line depicts CBH.**