# Peer review of "Understanding aerosol-cloud interactions in the development of orographic cumulus congestus during IPHEx"

_Atmospheric Chemistry and Physics, 2017_

## Referee Comment (RC1) · Anonymous Referee #1 · 12 Jun 2017

This paper shows some very interesting results by applying a modified cloud parcel model (CPM) to study the aerosol-cloud interactions (ACI). The authors first describe the processes included in the CPM, such as collision-coalescence, and lateral entrainment. Then the authors show the observations from the IPHEx field campaign, which include ground and aircraft measurements. A series of parameter sensitivity analyses are carried out using the CPM and the model outputs are compared with the observations. This paper is certainly of great interest and well within the scope of ACP, but it can also be significantly improved. I have several comments:

1. Page 2, Line 30-34: The authors try to point out the shortcomings of parameteri-

zation in the model. Instead of using "inadequate to capture the spatial and temporal resolution", it would be better to list some detailed discrepancies between model and observations from the literature.

2. Page 8, Line 10-13: Euler method is used as the integration method for the collision-coalescence processes. The reason is "to examine its role individually in cloud formation". Does this mean the collision-coalescence processes do not suffer from stiffness? How would you justify the benefit of using the Euler method while it may potentially cause numerical instability in the model?

3. Page 11, Line 10: in Fig. 5b, when CDP LWC value is close to zero, there is a clear intercept of ∼0.05 g m-3 in King LWC. As such, including an intercept value in the linear regression would produce a better fit (i.e., fit to the equation y = a x + b instead of y = a x). Please explain why the intercept is not included in the linear regression.

4. Page 15, Line 6-8: The underestimation of supersaturation by model is argued to be due to the uncertainties of temperature and humidity in WRF simulation. However, in the sensitivity test discussed in Appendix B1, adjusting the temperature and humidity increase the supersaturation to ∼0.5% (Fig. B1(a)), which is still significantly smaller than the observations. This indicates that the temperature and humidity in WRF simulation do not have a strong influence on the supersaturation profile. Could the authors list other factors that affect the supersaturation profile?

Minor comments:

1. Page 2, Line 25-26: the scale gap should be 5 to 9 orders of magnitude when comparing $\mu$m, cm with km.

2. Page 5, Line 11: "Fig. 2" appears earlier in the text than "Fig. 1" (Page 8, Line 17), thus the order of Fig.1 and Fig. 2 should be switched.

3. Page 8, Line 29: "Aerosol observations were collected" should be "Aerosol observations were carried out".

4. Page 8, Line 30: first time "MSL" appears, give full name.

5. Page 9, Line 1: "scanning mobility particle counter system (SMPS)". Please provide the manufacturer of the instrument. This applies to other instruments listed thereafter.

6. Page 9, Line 7: "shows very close agreement with the SMPS measurements". Maybe the authors could provide some data (e.g., correlation coefficient) to show the degree of agreement.

7. Page 9, Line 11: "8 mins" should be "8 min".

8. Page 14, Line 12: "range [0.001–1.0]" should be "range [0.001, 1.0]" or "range 0.001–1.0".

9. Page 15, Line 7: "obtained the WRF simulation" should be "obtained from the WRF simulation".

10. Page 34, Table 2: please make the significant figures consistent within each parameter.

11. Page 42, Fig 7c and 7d. It is difficult to differentiate lines in same color from each other. Please consider using different colors for each line if possible.

---

## Referee Comment (RC2) · Anonymous Referee #2 · 1 Jul 2017

This study attempts to reach closure between simulated and observed relationships among CCN, cloud base updraft, drop concentrations, cloud mixing extent with ambient air, supersaturation at cloud base and above it, and condensation accommodation coefficient. This is a worthwhile and ambitious goal, but unfortunately the available measurements are insufficient for achieving any unambiguous closure. The authors are aware of this and attempt to supplement the missing measurements with assumptions that cannot be supported. This undermines the basis of the validity of the claimed closure.

The specific problems with the basis of the study are:

[Figure]

1. Are these really cumulus congestus clouds? The nature of the clouds does not appear to be cumulus congestus, but rather deep precipitating clouds, at least by the radar data shown in Figure 2. Panel b shows an intense downdraft with Doppler folding velocity well exceeding 10 m/s. The reflectivity at that time, as shown in panel a, is so large that the radar echo at that time is fully attenuated above 3 to 4 km.

2. Is cloud base updraft 0.5 m/s? The cloud base updraft for convective cores should be taken as the peak and not average values. During the time before 12:31 they reach values between 1 and 2 m/s. Cloud base is where the green changes to white between 2.5 and 3 km MSL. The green indicates the fall velocity of the light precipitation from the clouds below their base.

3. The updraft vertical profile. The stable stratification implies that the updrafts were forced mainly by orography and/or gust front from the nearby strong downdraft. Therefore, a parcel model of convective updraft is hardly applicable for this situation.

4. CCN concentrations at cloud base. The CCN at cloud base is not measured, but rather assumed. It is assumed that the surface measured CCN decays exponentially with height with a scale height of 1 km. However, in convective clouds the boundary layer must be well mixed, with a constant aerosol mixing ratio. It must be so with updrafts rooted near the surface in the solar heated boundary layer, if we take as a valid the assumptions of the convective parcel.

5. The supersaturation in clouds. The authors report measured supersaturation (SS) in clouds. However, measuring SS in clouds is highly challenging, because in cloud temperature has to be measured at a very high accuracy as well as mixing ratio of water vapor. A major issue of measuring temperature in clouds is the effect of wetting of the temperature probe and the resultant evaporative cooling, which is far from being completely solved in reversal flow thermometer. To convince us that SS can be measured with a useful accuracy the authors have to provide the full description of the method of its calculation along with error calculation. They report SS=3% in a cloud

with drop concentration of about 400 cmˆ-3 and updraft speed of nearly 1 m/s. This is physically impossible. The SS in such conditions must be a fraction of 1%.

6. The cloud drop size distribution widened in places unrealistically for convective clouds, as in ECR (Figure 8a). I strongly suspect that the clouds had precipitation falling from above, or recirculating cloudy downdraft air which had already produced precipitation. This increases the question about the applicability of a parcel model for these clouds.

On top of these problems, there are many issues, including:

Page 14 line 27: The text reads: "Consequently, smaller aerosol particles with high concentrations are activated due to a higher Smax further up from the cloud base, resulting in a direct increase in cloud droplet numbers (Fig. 11c)." However, the parcel model shows a decreasing drop concentrations with height for all scenarios. There is no closure here. Furthermore, Smax is a property of cloud base, not well above it.

Page 14 lines 30-33: There will be always a value of accommodation coefficient that matches the observed drop concentration. It can only be constrained if both cloud base updraft and CCN are known, because an increase in cloud base updraft has a similar result as of decreasing the accommodation coefficient. The same ambiguity applies to increasing CCN vs. decreasing the accommodation coefficient. A closure cannot be possibly reached with such uncertainties with respect to both cloud base CCN and updraft speed.

Because of the problems with the basis of the paper as well as with its implementation, the conclusions of the paper cannot be supported.

---

## Author Comment (AC1) · 5 Sep 2017

We thank the reviewer for the comments, which were fully taken into consideration in the revised manuscript. Reviewer comments (in black) and our replies (in blue) are provided as a supplemental PDF document.

Please also note the supplement to this comment:

Please also note the supplement to this comment: https://www.atmos-chem-phys-discuss.net/acp-2017-396/acp-2017-396-AC1-supplement.zip